# Synteny-based analyses indicate that sequence divergence is not the main source of orphan genes

**Nikolaos Vakirlis[1], Anne-Ruxandra Carvunis[2]\*, Aoife McLysaght[1]\***

[1]Smurfit Institute of Genetics, Trinity College Dublin, University of Dublin, Dublin, Ireland; [2]Department of Computational and Systems Biology, Pittsburgh Center for Evolutionary Biology and Medicine, School of Medicine, University of Pittsburgh, Pittsburgh, United States

**Abstract** The origin of 'orphan' genes, species-specific sequences that lack detectable homologues, has remained mysterious since the dawn of the genomic era. There are two dominant explanations for orphan genes: complete sequence divergence from ancestral genes, such that homologues are not readily detectable; and de novo emergence from ancestral non-genic sequences, such that homologues genuinely do not exist. The relative contribution of the two processes remains unknown. Here, we harness the special circumstance of conserved synteny to estimate the contribution of complete divergence to the pool of orphan genes. By separately comparing yeast, fly and human genes to related taxa using conservative criteria, we find that complete divergence accounts, on average, for at most a third of eukaryotic orphan and taxonomically restricted genes. We observe that complete divergence occurs at a stable rate within a phylum but at different rates between phyla, and is frequently associated with gene shortening akin to pseudogenization.

**\*For correspondence:**
anc201@pitt.edu (A-RC);
Aoife.McLysaght@tcd.ie (AMcL)

**Competing interests:** The authors declare that no competing interests exist.

## Introduction

Extant genomes contain a large repertoire of protein-coding genes which can be grouped into families based on sequence similarity. Comparative genomics has heavily relied on grouping genes and proteins in this manner since the dawn of the genomic era (*Rubin, 2000*). Within the limitations of available similarity-detection methods, we thus define thousands of distinct gene families. Given that the genome and gene repertoire of the Last Universal Common Ancestor (LUCA) was likely small relative to that of most extant eukaryotic organisms (*Becerra et al., 2007*; *Goldman et al., 2013*) (*Figure 1A*), what processes gave rise to these distinct gene families? Answering this question is essential to understanding the structure of the gene/protein universe, its spectrum of possible functions, and the evolutionary forces that ultimately gave rise to the enormous diversity of life on earth.

To some extent, the distinction between gene families is operational and stems from our imperfect similarity-detection ability. But to a larger extent it is biologically meaningful because it captures shared evolutionary histories and, by extension, shared properties between genes that are useful to know (*Gabaldón and Koonin, 2013*; *Koonin, 2005*). Genes that cannot be assigned to any known gene family have historically been termed 'orphan'. This term can be generalized to Taxonomically Restricted Gene (TRG), which includes genes that belong to small families found only across a closely related group of species and nowhere else (*Wilson et al., 2005*).

By definition, orphan genes and TRGs can be the result of two processes. The first process is divergence of pre-existing genes (*Tautz and Domazet-Lošo, 2011*). Given enough time, a pair of genes that share a common ancestor (homologous genes) can reach the 'twilight zone' (*Doolittle, 1981*), a point at which similarity is no longer detectable. From a sequence-centric standpoint,

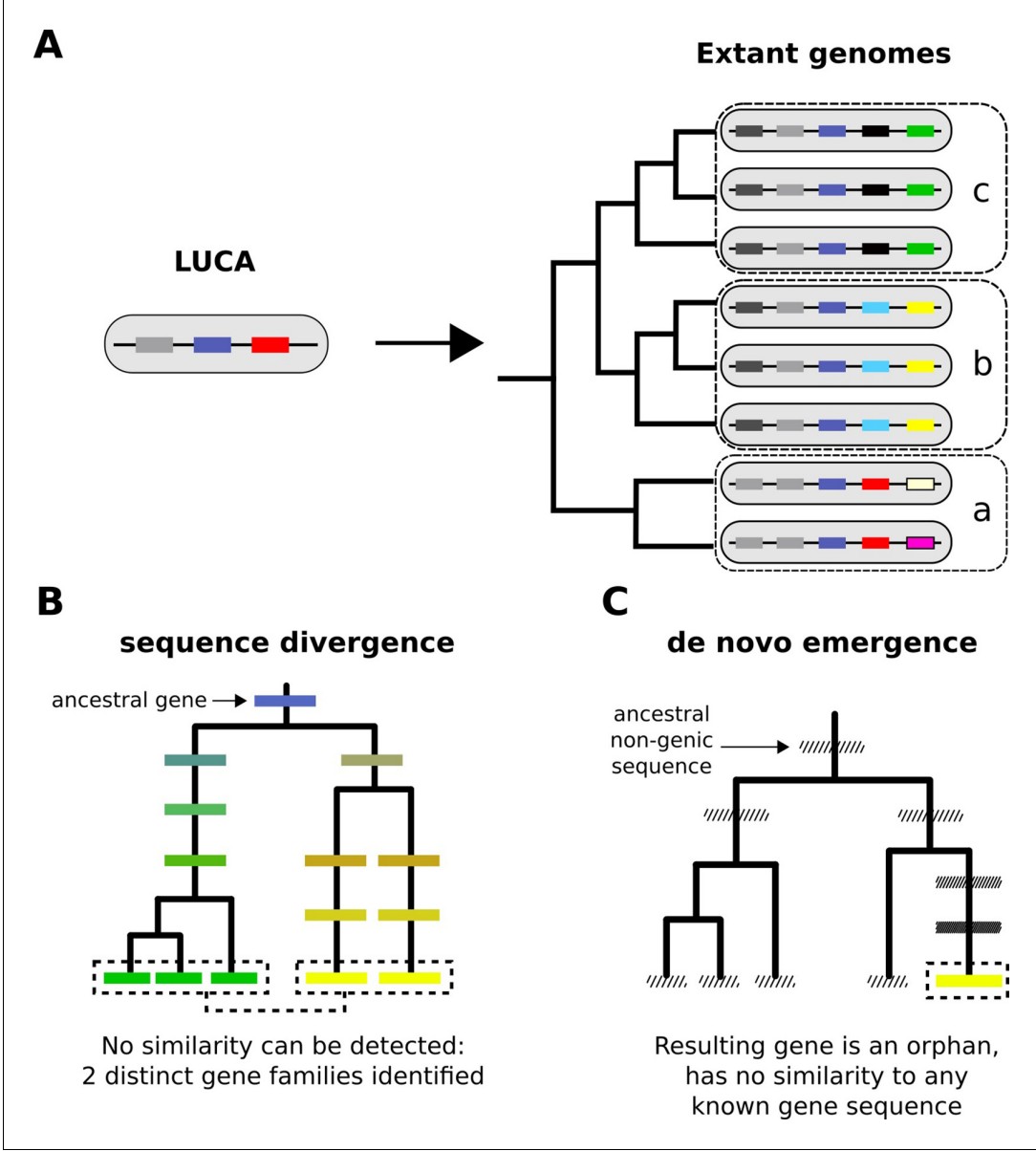

**Figure 1.** From a limited set of genes in LUCA to the multitudinous extant patterns of presence and absence of genes. (**A**) Cartoon representation of the LUCA gene repertoire and extant phylogenetic distribution of gene families (shown in different colours, same colour represents sequence similarity and homology). Dashed boxes denote different phylogenetic species groups. Light grey and dark blue gene families cover all genomes and can thus be traced back to the common ancestor. Other genes may have more restricted distributions; for example, the yellow gene is only found in group b, the black gene in group c. The phylogenetic distribution of gene family members allows us to propose hypotheses about the timing of origination of each family. (**B**) Sequence divergence can gradually erase all similarity between homologous sequences, eventually leading to their identification as distinct gene families. Note that divergence can also occur after a homologous gene was acquired by horizontal transfer. Solid boxes represent genes. Sequence divergence is symbolized by divergence in colour. (**C**) De novo emergence of a gene from a previously non-genic sequence along a specific lineage will almost always result in a unique sequence in that lineage (cases of convergent evolution can in theory occur). Hashed boxes represent non-genic sequences.

we can consider such entities as bearing no more similarity than expected by chance. They are the seeds of two new gene families (*Figure 1B*). An example of this was found when examining yeast duplicates resulting from whole genome duplication (WGD) where it was reported that about 5% of the ~500 identified paralogue pairs had very weak or no similarity at all (*Wolfe, 2004*). Divergence of pre-existing genes can occur during vertical descent (*Figure 1B*), as well as following horizontal transfer of genetic material between different species (*Dunning Hotopp, 2011*). The second process is de novo emergence from previously non-genic sequences (*Vakirlis et al., 2018*; *McLysaght and Guerzoni, 2015*; *Van Oss and Carvunis, 2019*) (*Figure 1C*). For a long time, divergence was considered to be the only realistic evolutionary explanation for the origin of new gene families (*Long et al., 2003*), while de novo emergence has only recently been appreciated as a widespread phenomenon (*Van Oss and Carvunis, 2019*; *Carvunis et al., 2012*; *Long et al., 2013*; *Tautz, 2014*). De novo emergence is thought to have a high potential to produce entirely unique genes (*Schlötterer, 2015*) (though examples of convergent selection exist, see *Baalsrud et al., 2018*; *Zhuang et al., 2019*), whereas divergence, being more gradual, can stop before this occurs. What is the relative contribution of these two mechanisms to the 'mystery of orphan genes' (*Dujon, 1996*)?

We set out to study the process of complete divergence of genes by delving into the 'unseen world of homologs' (*Wolfe, 2004*). More specifically, we sought to understand how frequently homologues diverge beyond recognition, reveal how the process unfolds, and explicitly identify resulting TRGs. To do so, we developed a novel synteny-based approach for homology detection and applied it to three lineages. Our approach allowed us to trace the limits of similarity searches in the context of homologue detection. We show that genes which diverge beyond these limits exist, that they are being generated at a steady rate during evolution, and that they account, on average, for at most a third of all genes without detectable homologues. All but a small percentage of these undetectable homologues lack similarity at the protein domain level. Finally, we study specific examples of novel genes that have originated or are on the verge of originating from pre-existing ones, revealing a possible role of gene disruption and truncation in this process. We show that in the human lineage, this evolutionary route has likely given rise to at least two mammal-specific, cancer-related genes.

## Results

### A synteny-based approach to establish homology beyond sequence similarity

To estimate the frequency at which homologues diverge beyond recognition, we developed a pipeline that allows the identification of candidate homologous genes regardless of whether pairwise sequence similarity can be detected. The central idea behind our pipeline is that genes found in conserved syntenic positions in a pair of genomes will usually share ancestry. The same basic principle has been previously used to detect pairs of WGD paralogues in yeast (*Wolfe and Shields, 1997*; *Kellis et al., 2004*; *Dietrich, 2004*) and more recently to identify homologous long non-coding RNAs (*Herrera-Úbeda et al., 2019*). Coupled with the knowledge that biological sequences diverge over time, this allows us to estimate how often a pair of homologous genes will diverge beyond detectable sequence similarity in the context of syntenic regions. This estimate can then be extrapolated genome-wide to approximate the extent of origin by complete divergence for orphan genes and TRGs outside of syntenic regions, provided that genes outside regions of conserved synteny have similar evolutionary rates as genes inside syntenic regions. The estimates that we will provide of the rate of divergence beyond recognition inside synteny blocks are best viewed as an upper-bound of the true rate because some of the genes found in conserved syntenic positions in a pair of genomes will not be homologous. If we could remove all such cases, the rate of divergence beyond recognition would only decrease, but not increase, relative to our estimate (*Figure 2A*).

*Figure 2B* illustrates the main steps of the pipeline and the full details can be found in Materials and methods. Briefly, we first select a set of target genomes to compare to our focal genome (*Figure 2B*, step 1). Using precomputed pairs of homologous genes (those belonging to the same OrthoDB [*Kriventseva et al., 2008*] group) we identify regions of conserved micro-synteny. Our operational definition of conserved micro-synteny consists of cases where a gene in the focal genome is found within a conserved chromosomal block of at least three genes: the immediate

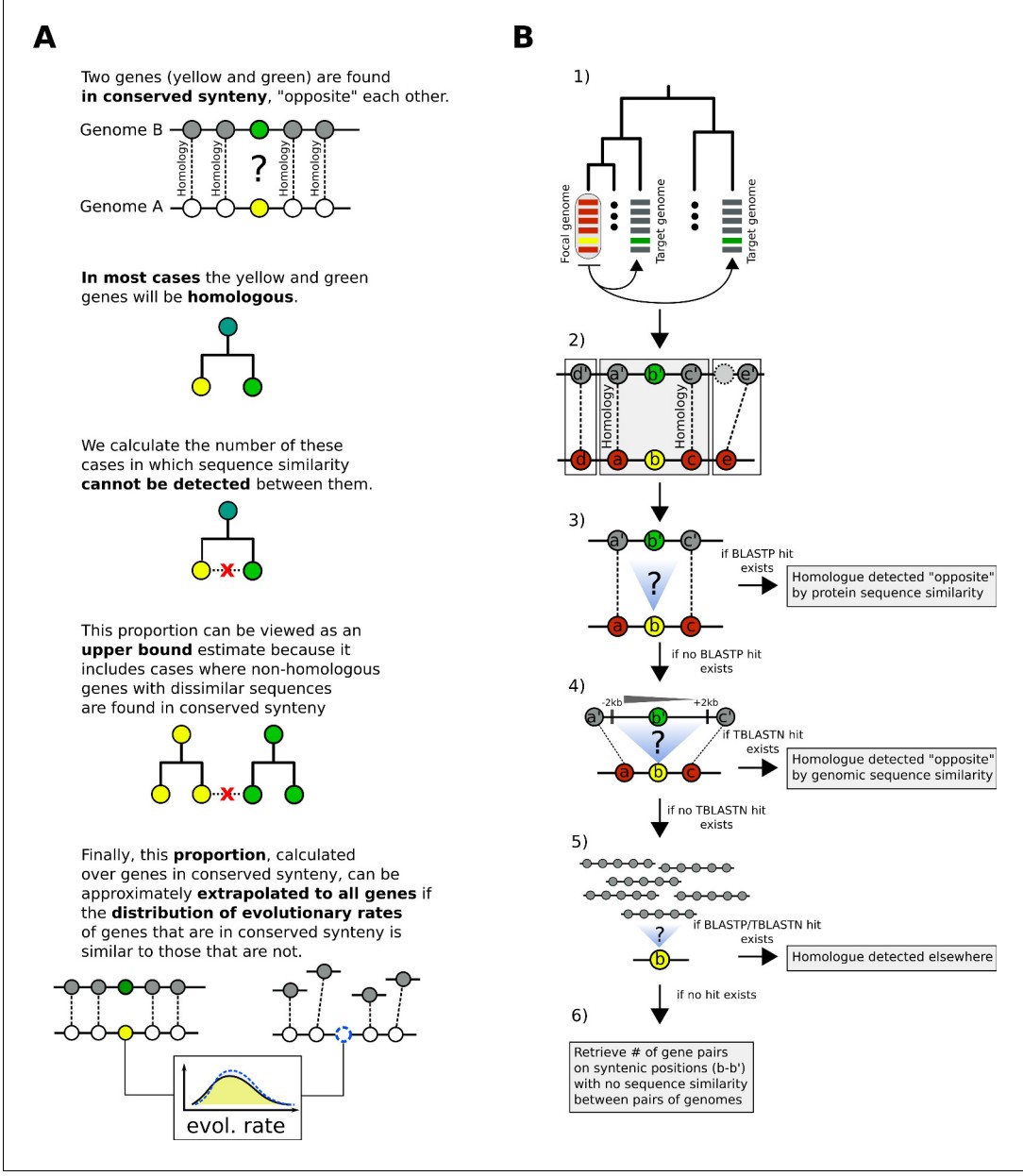

**Figure 2.** Summary of the main concept and pipeline of identification of putative homologous pairs with undetectable similarity between pairs of genomes. (**A**) Summary of the reasoning we use to estimate the proportion of genes in a genome that have diverged beyond recognition. (**B**) Pipeline of identification of putative homologous pairs with undetectable similarity. 1) Choose focal and target species. Parse gene order and retrieve homologous relationships from OrthoDB for each focal-target pair. Search for sequence similarity by BLASTP between focal and target proteomes, one target proteome at a time. 2) For every focal gene (*b*), identify whether a region of conserved micro-synteny exists, that is when the upstream (*a*) and downstream (*c*) neighbours have homologues (*a'*, *c'*) separated by either one or two genes. This conserved micro-synteny allows us to assume that *b* and *b'* are most likely homologues. Only cases for which the conserved micro-synteny region can be expanded by one additional gene are retained. Specifically, if genes *d* and *e* have homologues, these must be separated by at most one gene from *a'* and *c'*, respectively. A per-species histogram of the number of genes with at least one identified region of conserved micro-synteny can be found in *Figure 2—figure supplement 1*. For all genes where at least one such configuration is found, move to the next step. 3) Check whether a precalculated BLASTP hit exists (by our proteome searches) between query (*b*) and candidate homologue (*b'*) for a given E-value threshold. If no hit exists, move to the next step. 4) Use TBLASTN to search for similarity between the query (*b*) and the genomic region of the conserved micro-synteny (-/+ 2 kb around the candidate homologue gene) for a

*Figure 2 continued on next page*

*Figure 2 continued*

given E-value threshold. If no hit exists, move to the next step. 5) Extend the search to the entire proteome and genome. If no hit exists, move to the next step. 6) Record all relevant information about the pairs of sequences forming the $b - b'$ pairs of step 2. Any statistically significant hit at steps 3–5 is counted as detected homology by sequence similarity. In the end, we count the total numbers of genes in conserved micro-synteny without any similarity for each pair of genomes.

The online version of this article includes the following figure supplement(s) for figure 2:

**Figure supplement 1.** Total number of genes in the focal species genome for which a region in conserved micro-synteny was identified in a given target species (x axis).

---

downstream and upstream neighbours of the focal gene must have homologues in the target genome that are themselves separated by at most one or two genes and, if the genes immediately next to these neighbours (second neighbours of the focal gene) have homologues in the target genome, these must also be separated from the homologues of the immediate neighbours by at most one gene (*Figure 2B*, step 2; see Materials and methods for details). Since the choice of synteny criterion can have an impact on downstream analyses we have also used one more relaxed and one more stringent definition (see Materials and methods; results using these alternative definitions are presented later). All focal genes for which at least one region of conserved micro-synteny, in any target genome, is identified, are retained for further analysis. This step establishes a list of focal genes with at least one presumed homologue in one or more target genomes (i.e., the gene located in the conserved location in the micro-synteny block).

We then examine whether the focal gene has any sequence similarity in the target species. We search for sequence similarity in two ways: comparison with annotated genes (proteome), and comparison with the genomic DNA (genome). First, we search within BLASTP matches that we have pre-computed ourselves (these are different from the OrthoDB data) using the complete proteome of the focal species as query against the complete proteome of the target species. Within this BLASTP output we look for matches between the query gene and the candidate gene (that is, between $b$ and $b'$, *Figure 2B*, step 3). If none is found then we use TBLASTN to search the genomic region around the candidate gene $b'$ for similarity to the query gene $b$ (*Figure 2B*, step 4, see figure legend for details). If no similarity is found, the search is extended to the rest of the target proteome and genome (*Figure 2B*, step 5). If there is no sequence similarity after these successive searches, then we infer that the sequence has diverged beyond recognition. After having recorded whether similarity can be detected for all eligible query genes, we finally retrieve the focal-target pairs and produce the found and not found proportions for each pair of genomes.

We applied this pipeline to three independent datasets using as focal species *Saccharomyces cerevisiae* (yeast), *Drosophila melanogaster* (fly) and *Homo sapiens* (human). We included 17, 16 and 15 target species, respectively, selected to represent a wide range of evolutionary distances from each focal species (see Materials and methods). The numbers of cases of conserved micro-synteny detected for each focal-target genome pair is shown in *Figure 2—figure supplement 1*.

## Selecting optimal BLAST E-value cut-offs

Homology detection is highly sensitive to the technical choices made during sequence similarity searches (*Tautz and Domazet-Lošo, 2011*; *Arendsee et al., 2019*). We therefore sought to explore how the choice of E-value threshold would impact interpretations of divergence beyond similarity. First, we performed BLASTP searches of the focal species' total protein sequences against the total reversed protein sequences of each target species. Matches produced in these searches can safely be considered 'false homologies' since biological sequences do not evolve by reversal (*Frith, 2011*) (see Materials and methods). These false homologies were then compared to 'undetectable homologies': cases with conserved micro-synteny (presumed homologues) but without any detectable sequence similarity.

In *Figure 3A*, we can see how the ratios of undetectable and false homologies vary as a function of the BLAST E-value threshold used. The proportion of undetectable homologies depended quasi-linearly on the E-value cut-off. By contrast, false homologies depended exponentially on the cut-off, as expected from the E-value definition. Furthermore, the impact of E-value cut-off was more

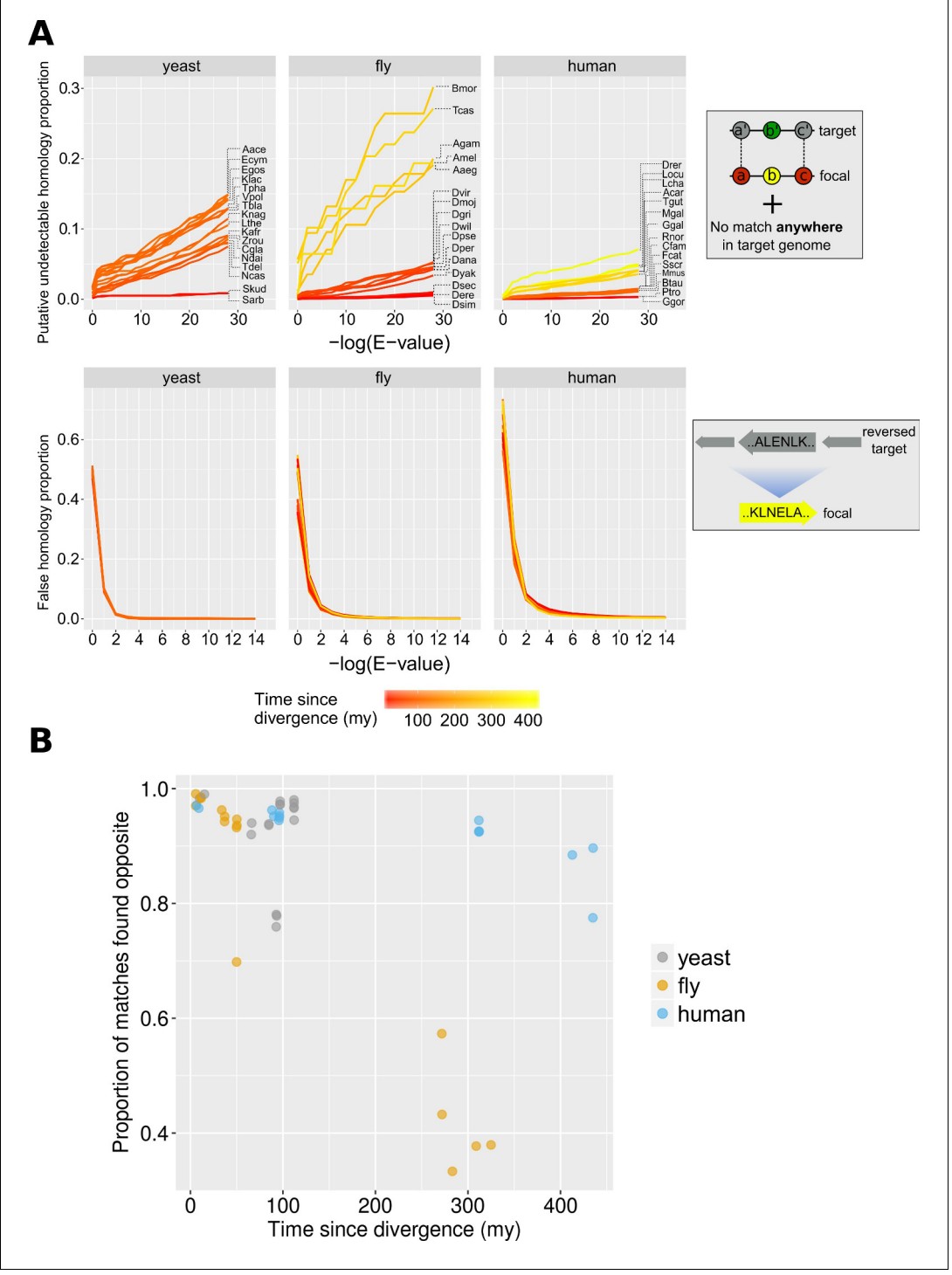

**Figure 3.** Proportions of false and undetectable homologies for a range of E-value cut-offs. (**A**) Proportions of false and undetectable homologies as a function of the E-value cut-off used. Abbreviations of species names can be found in **Table 1**. Putative undetectable homology proportion (top row) is defined as the percentage of all genes with at least one identified region of conserved micro-synteny (and thus likely to have a homologue in the target genome) that have no significant match anywhere in the target genome (see Materials and methods and **Figure 2**). False homology proportion (bottom row) is defined as a significant match to the reversed proteome of the target species (see Materials and methods). Divergence time estimates were obtained from www.TimeTree. org. Data for this figure can be found in **Figure 3—source data 2** (upper plots) and **Figure 3—source data 3** (lower plots). (**B**) Proportion (out of all genes with sequence matches) where a match is found in the predicted

*Figure 3 continued on next page*

*Figure 3 continued*

region ('opposite') in the target genome for the three datasets, using the relaxed E-value cut-offs (0.01, 0.01, 0.001 for yeast, fly and human, respectively [$10^{-4}$ for comparison with chimpanzee]), as a function of time since divergence from the respective focal species. Data can be found in *Figure 3—source data 1*.

The online version of this article includes the following source data for figure 3:

**Source data 1.** Data from focal-target genome comparisons.
**Source data 2.** Data on undetectable homologies for different E-value cut-offs used to generate the top panel of *Figure 3A*.
**Source data 3.** Data on false homologies for different E-value cut-offs used to generate the bottom panel of *Figure 3A*.

---

pronounced in comparisons of species separated by longer evolutionary distances, whereas it was almost non-existent for comparisons amongst the most closely related species. Conversely, there seems to be no dependence between percentage of false homologies and evolutionary time across the range of E-values that we have tested (all lines overlap in the graphs in the bottom panel of *Figure 3A*). This means that, when comparing relatively closely related species, failing to appropriately control for false homologies would have an overall more severe effect on homology detection than failing to account for false negatives.

In the context of phylostratigraphy (estimation of phylogenetic branch of origin of a gene based on its taxonomic distribution; *Domazet-Loso et al., 2007*), gene age underestimation due to BLAST 'false negatives' has been considered a serious issue (*Moyers and Zhang, 2014*), although the importance of spurious BLAST hits generating false positives has also been stressed (*Domazet-Lošo et al., 2017*). We defined a set of E-value cut-offs optimised for phylostratigraphy, by choosing the highest E-value that keeps false homologies under 5%. This strategy emphasizes sensitivity over specificity. We have also calculated general-use optimal E-values by using a balanced binary classification measure (see Materials and methods). The phylostratigraphy optimal E-value thresholds are 0.01 for all comparisons using yeast and fly as focal species and 0.001 for those of human, except for chimpanzee ($10^{-4}$). These are close to previously estimated optimal E-value cut-offs for

**Table 1.** Names and abbreviations of target species included in the three datasets.

| Full name | Abbr. | Full name | Abbr. | Full name | Abbr. |
|---|---|---|---|---|---|
| *Saccharomyces kudriavzevii* | Skud | *Drosophila sechellia* | Dsec | *Pan troglodytes* | Ptro |
| *Saccharomyces arboricola* | Sarb | *Drosophila simulans* | Dsim | *Gorilla gorilla* | Ggor |
| *Naumovozyma castellii* | Ncas | *Drosophila erecta* | Dere | *Mus musculus* | Mmus |
| *Naumovozyma dairenensis* | Ndai | *Drosophila yakuba* | Dyak | *Rattus norvegicus* | Rnor |
| *Kazachstania naganishii* | Knag | *Drosophila ananassae* | Dana | *Bos taurus* | Btau |
| *Kazachstania africana* | Kafr | *Drosophila persimilis* | Dper | *Canis familiaris* | Cfam |
| *Vanderwaltozyma polyspora* | Vpol | *Drosophila pseudoobscura* | Dpse | *Felis catus* | Fcat |
| *Tetrapisispora blattae* | Tbla | *Drosophila mojavensis* | Dmoj | *Sus scrofa* | Sscr |
| *Tetrapisispora phaffii* | Tpha | *Drosophila willistoni* | Dwil | *Anolis carolinensis* | Acar |
| *Torulaspora delbrueckii* | Tdel | *Drosophila grimshawi* | Dgri | *Gallus gallus* | Ggal |
| *Candida glabrata* | Cgla | *Drosophila virilis* | Dvir | *Meleagris gallopavo* | Mgal |
| *Zygosaccharomyces rouxii* | Zrou | *Anopheles gambiae* | Agam | *Taeniopygia guttata* | Tgut |
| *Kluyveromyces lactis* | Klac | *Aedes aegypti* | Aaeg | *Latimeria chalumnae* | Lcha |
| *Lachancea thermotolerans* | Lthe | *Bombyx mori* | Bmor | *Danio rerio* | Drer |
| *Eremothecium cymbalariae* | Ecym | *Tribolium castaneum* | Tcas | *Lepisosteus oculatus* | Locu |
| *Ashbya aceri* | Aace | *Apis mellifera* | Amel | | |
| *Eremothecium gossypii* | Egos | | | | |

identifying orphan genes in *Drosophila*, found in the range of $10^{-3}$ - $10^{-5}$, see ref (***Domazet-Loso and Tautz, 2003***). These cut-offs have been used for all downstream analyses.

We find that, for the vast majority of focal genes examined that do have matches, the match occurs in the predicted region ('opposite'), that is within the region of conserved micro-synteny. In 36/48 pair-wise species comparisons, at least 90% of the focal genes in micro-synteny for which at least one match was eventually found in the target genome, a match was within the predicted micro-syntenic region (***Figure 3B***). This finding supports the soundness of our synteny-based approach for homologue identification.

In total, we were able to identify 180, 81 and 156 unique focal species genes in the dataset of yeast, fly and human respectively, that have at least one undetectable homologue in at least one target species but no significant sequence similarity to that homologue or to any other part of the target genome (see ***Figure 4—figure supplement 1*** for two exemplars of these findings).

## The rate of 'divergence beyond recognition' and its contribution to the total pool of genes without similarity

How quickly do homologous genes become undetectable? In other words, given a pair of genomes from species separated by a certain amount of evolutionary time, what percentage of their genes will have diverged beyond recognition? Within phyla, the proportion of putative undetectable homologues correlated strongly with time since divergence, suggesting a continuous process acting during evolution (***Figure 4***). However, different rates were observed between phyla, represented by the slopes of the fitted linear models in ***Figure 4***. Genes appeared to be diverging beyond recognition at a faster pace in the yeast and fly lineages than in the human lineage.

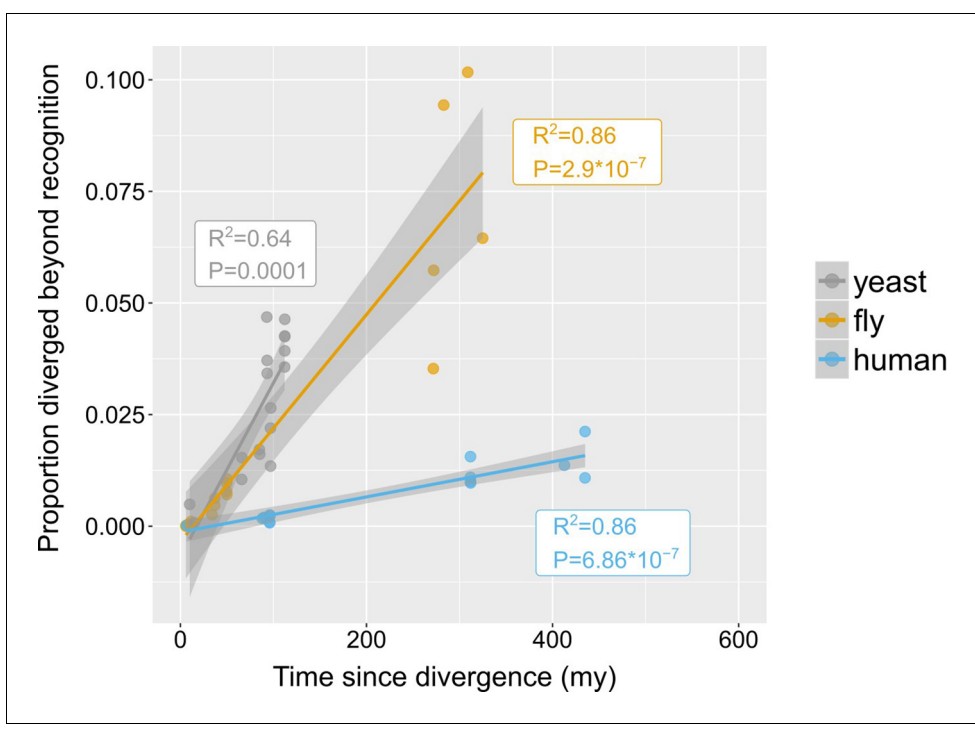

**Figure 4.** Rates of divergence beyond recognition. Putative undetectable homology proportion in focal - target species pairs plotted against time since divergence of species. The y axis represents the proportion of focal genes in micro-synteny regions for which a homologue cannot be detected by similarity searches in the target species. Linear fit significance is shown in the graph. Points have been jittered along the X axis for visibility. Two exemplars of focal-target undetectable homologues can be found in ***Figure 4—figure supplement 1***. Data can be found in ***Figure 3—source data 1***.

The online version of this article includes the following figure supplement(s) for figure 4:

**Figure supplement 1.** Two examples of putative homologues diverged beyond recognition.

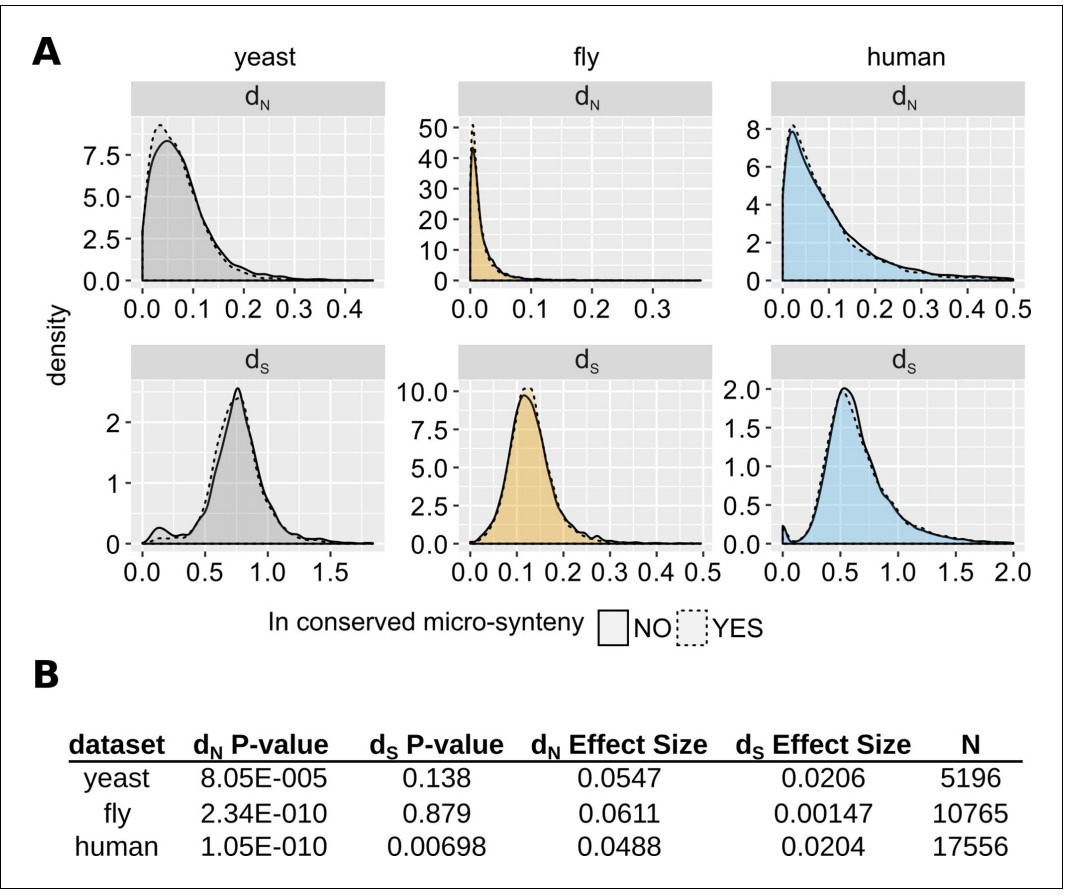

Figure 5. Comparison of evolutionary rates between genes inside and outside conserved micro-synteny regions. (A) Density plots of $d_S$ and $d_N$ distributions. Outliers are not shown for visual purposes Data can be found in *Figure 5—source data 1*. (B) Statistics of unpaired Wilcoxon test comparisons between genes inside and outside of conserved micro-synteny. Effect size was calculated using Rosenthal's formula (*Rosenthal et al., 1994*) (Z/sqrt (N)).

The online version of this article includes the following source data for figure 5:

**Source data 1.** $d_N$ and $d_S$ data used to generate *Figure 5* and the accompanying statistics.

We next sought to estimate how much the process of divergence beyond recognition contributes to the genome-wide pool of genes without detectable similarity. To do so, we need to assume that the proportion of genes that have diverged beyond recognition in micro-synteny blocks (*Figure 4*) can be used as a proxy for the genome-wide rate of origin-by-divergence for genes without detectable similarity, irrespective of the presence of micro-synteny conservation. This in turn depends on the distribution of evolutionary rates inside and outside micro-synteny blocks.

We calculated the non-synonymous ($d_N$) and synonymous ($d_S$) substitution rates of genes found inside and outside regions of conserved micro-synteny relative to closely related species (Materials and methods). *Figure 5A* shows density plots of the distributions. The distributions of $d_S$ are statistically indistinguishable for genes inside and outside of micro-synteny regions in the yeast and fly datasets. The distributions of $d_N$ for all three datasets and $d_S$ for the human dataset show a statistically significant increase in genes outside conserved micro-synteny regions compared to genes inside such regions, but the effect size is minimal, almost negligible (Rosenthal's $R \sim 0.05$, *Figure 5B*). It is impossible to directly compare the evolutionary rates of genes lacking homologues inside and outside conserved micro-synteny. However, such genes only account for a miniscule percentage of all genes in the genome: 0.0013, 0.008 and 0.029 in fly, human and yeast respectively. Despite these minimal caveats, evolutionary rates are globally very similar inside and outside regions of conserved micro-synteny, allowing to extrapolate with confidence.

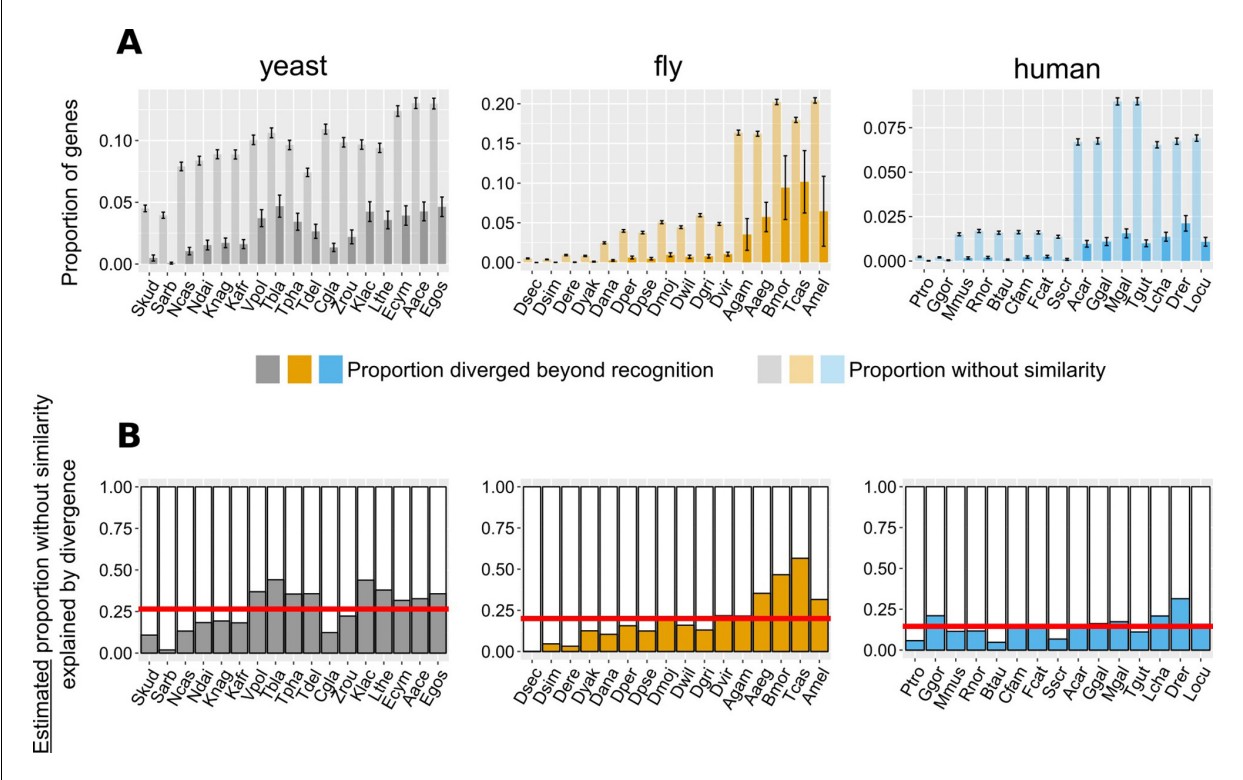

**Figure 6.** Contribution of divergence beyond recognition to observed numbers of genes without detectable similarity. (**A**) Proportion of genes with undetectable homologues in micro-synteny regions (thus likely diverged beyond recognition, solid bars) and proportion of total genes without similarity, genome-wide (transparent bars), in the different focal - target genome pairs. Schematic representation for how these proportions are calculated can be found in *Figure 6—figure supplement 1*. Error bars show the standard error of the proportion. (**B**) Estimated proportions of genes with putative undetectable homologues (explained by divergence) out of the total number of genes without similarity genome-wide. This proportion corresponds to the ratio of the micro-synteny proportion (solid bars in top panel) extrapolated to all genes, to the proportion calculated over all genes (transparent bars in top panel). See text for details. Red horizontal lines show averages. Species are ordered in ascending time since divergence from the focal species. Abbreviations used can be found in *Table 1*. The equivalent results using the phylogeny-based approach can be found in *Figure 6— figure supplement 3*. The impact of more/less stringent conserved synteny definitions on this result can be found in *Figure 6—figure supplement 2* (see also Materials and methods for details). Data for this figure and for *Figure 6—figure supplement 3* can be found in *Figure 6—source data 1*. The online version of this article includes the following source data and figure supplement(s) for figure 6:

**Source data 1.** Excel file with one dataset per sheet, containing the similarity and micro-synteny conservation information for every focal - target species comparison.
**Figure supplement 1.** Schematic representation of a toy example as an aid to understand how the proportion of genes without similarity that is explained by divergence is estimated.
**Figure supplement 2.** Impact of the definition of conserved micro-synteny to proportion explained by divergence.
**Figure supplement 3.** Phylogeny-based approach to estimate the contribution of divergence to TRGs.

We extrapolated the proportion of genes without detectable similarity that have originated by complete divergence, as calculated from conserved micro-synteny blocks (*Figure 4*), to all genes without similarity in the genome (*Figure 6*, see Materials and methods and *Figure 6—figure supplement 1* for detailed description). We found that, in most pairwise species comparisons, the observed proportion of all genes without similarity far exceeds that estimated to have originated by divergence (*Figure 6A*). The estimated contribution of divergence ranges from 0% in the case of *D. sechellia* (fly dataset), to 57% in the case of *T. castaneum* (fly dataset), with an overall average of 20.6% (*Figure 6B*).

The criteria used to define conserved micro-syntenic regions affect the number of regions identified (*Figure 6—figure supplement 2A*; Materials and methods) and can thus impact the proportions explained by divergence that are calculated based on such regions. Using a more relaxed conserved synteny definition (only one syntenic homologue on either side of the focal gene) had limited impact

on these results (overall average of 23% explained by divergence; *Figure 6—figure supplement 2B*; Materials and methods). We also used a more stringent definition of conserved micro-syntenic regions (an additional syntenic homologue on either side), but the number of regions identified was too low to extract meaningful conclusions with the more distant species pairs in our data sets (e.g. <5 for comparisons to all non-*Drosophila* species in the fly dataset, *Figure 6—figure supplement 2A*). In the species where the number of identified regions was sufficient, the average proportion explained by divergence dropped from 16% (using our current definition in this subset of species) to 10.9% (see *Figure 6—figure supplement 2B*; Materials and methods). The more pronounced difference in the stringent-current comparison could be explained by the fact that these proportions are calculated as the ratio of two small ratios (see *Figure 6—figure supplement 1*), which can be sensitive as the small ratio in the numerator changes with the conserved synteny definition. An alternative explanation is that as the synteny criterion becomes stricter, genes that satisfy it could be under stronger evolutionary constraints and therefore less likely to diverge. For both reasons, we recommend avoiding the application of stringent conserved synteny criteria when undertaking similar analyses, although ultimately the choice depends on the evolutionary distance and the more general degree of synteny conservation between the species being compared.

We also applied the same reasoning to estimate how much divergence beyond recognition contributes to TRGs. To this aim we calculated the fraction of focal genes lacking detectable homologues in a phylogeny-based manner, in the target species and in all species more distantly related to the focal species than the target species (see Materials and methods and *Figure 6—figure supplement 3A* for a schematic explanation). Again, the observed proportion of TRGs far exceeded that estimated to have originated by divergence (the contribution of divergence ranging from 0% to 52% corresponding to the first and before-last 'phylostratum' of the fly dataset tree respectively, with an overall average of 30%; *Figure 6—figure supplement 3B C*). Changing the conserved synteny definition impacted the proportions of TRGs explained by divergence similarly to what was seen in the pairwise case (*Figure 6—figure supplement 2B*). We estimate that the proportion of TRGs which originated by divergence-beyond-recognition, at the level of Saccharomyces, melanogaster subgroup, and primates are at most 45%, 20% and 24%, respectively (Materials and methods). Thus, we conclude that the origin of most genes without similarity cannot be attributed to divergence beyond recognition. This implies a substantial role for other evolutionary mechanisms such as de novo emergence and horizontal gene transfer, although horizontal gene transfer is not known to be frequent in metazoa.

## Properties of genes diverged beyond recognition

Even as homologous primary sequences diverge beyond recognition, it is conceivable that other ancestral similarities persist. We found weak but significant correlations between pairs of undetectable homologues in the human dataset when comparing G+C content (Spearman's rho = 0.25, p-value=$2\times10^{-5}$) and CDS length (Spearman's rho = 0.35, p-value=$1.5\times10^{-9}$). We also compared protein properties between the pairs of genes and found weak conservation for solvent accessibility, coiled regions and alpha helices only (yeast: % residues in solvent-exposed regions, rho = 0.14, p-value=0.0037; yeast and human: % residues in coiled protein regions, rho = 0.19, p-value=$8.25\times10^{-05}$ and rho = 0.14, p-value=0.017; human: % residues in alpha helices, rho = 0.2, p-value=0.00056).

We searched for shared Pfam (*Finn et al., 2016*) domains (protein functional motifs) and found that, in the yeast and human dataset, focal proteins had significantly fewer Pfam matches than their undetectable homologues (*Figure 7A*). Overall, a common Pfam match between undetectable homologues was found only for 12 pairs out of a total of 845 that we examined (1.4%). We also identified 13 additional cases of undetectable homologue pairs that, despite not sharing any pairwise similarity, belonged to the same OrthoDB group. Nonetheless, and despite the small sample size, genes forming these 25 pairs (corresponding to 17 distinct focal genes) were strongly correlated across 9 out of 10 features tested (Bonferroni-corrected P-values of <0.05; see *Figure 7B* and *Figure 7—source data 1*). Though rare, such cases of retention of similarity at the protein domain level, suggest the possibility of conservation of ancestral functional signals in the absence of sequence similarity.

One of these rare cases is *MNE1*, a 1992nt long *S. cerevisiae* gene encoding a protein that is a component of the mitochondrial splicing apparatus (*Watts et al., 2011*). The surrounding micro-

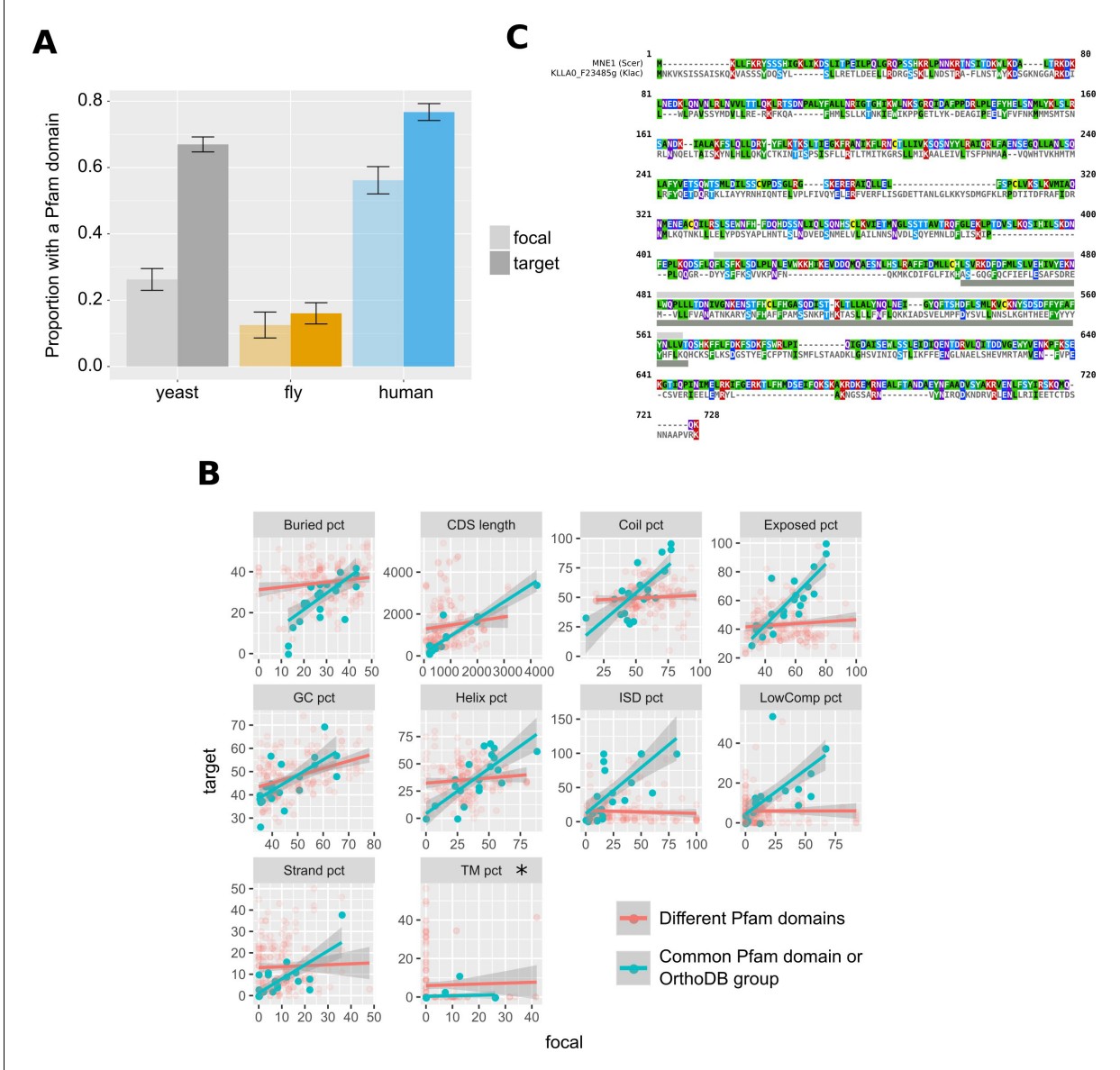

**Figure 7.** Pfam domains and other protein properties across undetectable homologue pairs. (**A**) Pfam domain matches in undetectable homologues. 'focal' (transparent bars) corresponds to the genes in the focal species, while 'target' (solid bars) to their putative undetectable homologues in the target species. Whiskers show the standard error of the proportion. The yeast comparison is statistically significant at p-value<$2.2\times10^{-16}$ and the human comparison at p-value=$2\times10^{-5}$ (Pearson's Chi-squared test). Raw numbers can be found in *Figure 7—source data 2*. (**B**) Distributions of properties of focal genes ('focal') and their undetectable homologues ('target'), when both have a significant match (p-value<0.001) to a Pfam domain or are members of the same OrthoDB group (blue points; n = 25), and when they lack a common Pfam match but both have at least one (red points; n = 183). All blue points correlations are statistically significant (Spearman's correlation, p-value<0.05; Bonferroni corrected) except from percentage of transmembrane residues (TM pct), marked with an asterisk. Details of correlations can be found in *Figure 7—source data 1*. All units are in percentage of residues, apart from 'GC pct' (nucleotide percentage) and CDS length (nucleotides). 'Buried pct': percentage of residues in regions with low solvent accessibility; 'CDS length': length of the CDS; 'Coil pct': percentage of residues in coiled regions; 'Exposed pct': percentage of residues in regions with high solvent accessibility; 'GC pct': Guanine Cytosine content; 'Helix pct': percentage of residues in alpha helices; 'ISD pct': percentage of residues in disordered regions; 'LowComp pct': percentage of residues in low complexity regions; 'Strand pct': percentage of residues in beta strands; 'TM pct': percentage of residues in transmembrane domains. Data can be found in *Figure 7—source data 3*. (**C**) Protein sequence alignment generated by MAFFT of *MNE1* and its homologue in *K. lactis*. Pfam match location is shown with a light grey rectangle in *S. cerevisiae*, and a dark grey one in *K. lactis*.

The online version of this article includes the following source data for figure 7:

**Source data 1.** Correlations of different protein properties between undetectable homologues.
**Source data 2.** Numbers of focal and target genes with Pfam matches and total numbers.
**Source data 3.** Data on common Pfam matches and gene/protein properties used to generate *Figure 7*.

synteny is conserved in five yeast species, and the distance from the upstream to the downstream neighbour is well conserved in all five (minimum of 2062nt and a maximum of 2379nt). In four of the five species the homologue can also be identified by sequence similarity, but *MNE1* of *S. cerevisiae* has no detectable protein or genomic similarity to its homologous gene in *Kluyveromyces lactis*,

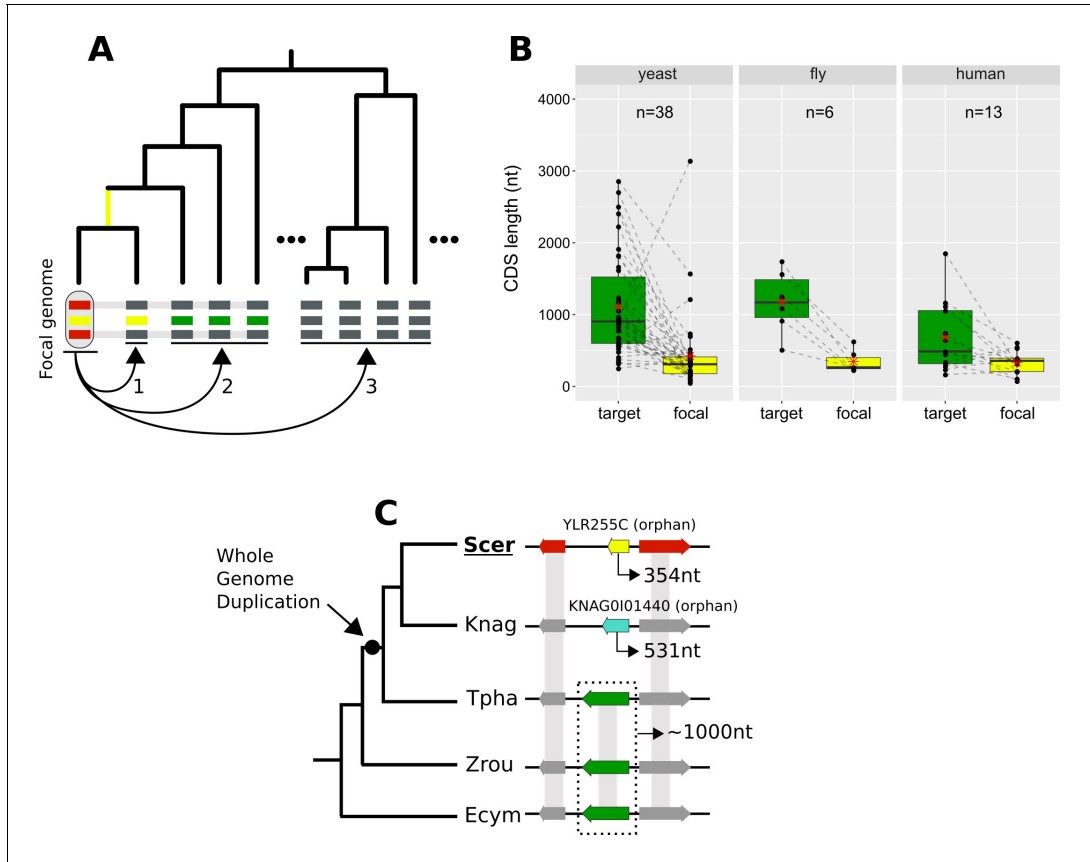

**Figure 8.** Lineage-specific divergence and gene length. (A) Schematic representation of the criteria used to detect lineage-specific divergence. 1, identification of any lineages where a homologue with a similar sequence can be detected (example for one lineage shown). 2, identification of at least two non-monophyletic target species with an undetectable homologue. 3, search in proteomes of outgroup species to ensure that no other detectable homologue exists. The loss of similarity can then be parsimoniously inferred as having taken place, through divergence, approximately at the common ancestor of the yellow-coloured genes (yellow branch). Leftmost yellow box: focal gene; Red boxes: neighbouring genes used to establish conserved micro-synteny; Green boxes: undetectable homologues. Grey bands connecting genes represent homology identifiable from sequence similarity. (B) CDS length distributions of focal genes and their corresponding undetectable homologues (averaged across all undetectable homologous genes of each focal one) in the three datasets. Dashed lines connect the pairs. All comparisons are statistically significant at p-value<0.05 (Paired Student's t-Test P-values: $2.5 \times 10^{-5}$, 0.0037, 0.03 in yeast, fly and human respectively). Distribution means are shown as red stars. Box colours correspond to coloured boxes representing genes in A, but only the focal genome gene (leftmost yellow gene in A) is included in the 'focal' category. Data can be found in *Figure 8—source data 1*. All focal-target undetectable homologue pairs (not just the ones included in this figure) can be found in *Figure 8—source data 2*. (C) Schematic representation of the species topology of 5 yeast species (see *Table 1* for abbreviations) and the genic arrangements at the syntenic region of *YLR255C* (shown at the 'Scer' leaf). Colours of boxes correspond to A. Gene orientations and CDS lengths are shown. The Whole Genome Duplication branch is tagged with a black dot. Genes grouped within dotted rectangles share sequence similarity with each other but not with other genes shown. Grey bands connecting genes represent homology identifiable from sequence similarity. Green genes: TPHA0B03620, ZYRO0E05390g, Ecym_2731.

The online version of this article includes the following source data for figure 8:

**Source data 1.** CDS lengths of focal genes and their undetectable homologues (averages), resulting from lineage-specific divergence.

**Source data 2.** All CDS and protein properties for all undetectable homologue pairs.

KLLA0_F23485g. Both the conserved micro-synteny and lack of sequence similarity are confirmed by examination of the Yeast Gene Order Browser (*Byrne and Wolfe, 2005*). Despite the lack of primary sequence similarity, the *S. cerevisiae* and *K. lactis* genes share a significant (E-value <0.001) Pfam match (Pfam accession PF13762.5; *Figure 7C*) and are members of the same fast-evolving OrthoDB group (EOG092E0K2I). The two are also not statistically different in terms of the protein properties that we calculated (Paired t-test p-value=0.8). Thus, *MNE1* exemplifies possible retention of ancestral properties in the absence of detectable pairwise sequence similarity.

## Lineage-specific gene origination through divergence

We looked for cases of focal genes that resulted from complete lineage-specific divergence along a specific phylogenetic branch (*Figure 8A*). When comparing the CDS lengths of these focal genes to those of their undetectable homologues, we found that focal genes tend to be much shorter (*Figure 8B*). This finding could partially explain the shorter lengths frequently associated with young genes (*Vakirlis et al., 2018*; *Carvunis et al., 2012*; *Wilson et al., 2017*; *Ruiz-Orera et al., 2015*). Through a lineage-specific shift of selection pressure, truncation of the gene could initiate accelerated divergence in a process that may at first resemble pseudogenization.

We sought a well-defined example to illustrate this process. *YLR255C* is a 354nt long, uncharacterized yeast ORF that is conserved across *S. cerevisiae* strains according to the Saccharomyces Genome Database (*Cherry et al., 2012*) (SGD). *YLR255C* is a species-specific, orphan gene. Our analyses identified undetectable homologues in four other yeast species. Three of them share sequence similarity with each other while the fourth one is another orphan gene, specific to *K. naganishii* (*Figure 8C*). The presence of two orphan genes in conserved synteny is strong evidence for extensive sequence divergence as an explanation of their origin. Based on the phylogenetic relationships of the species and the CDS lengths of the undetectable homologues, we can infer that the ancestor of *YLR255C* was longer (*Figure 8C*). Furthermore, given that *S. cerevisiae* and *K. naganishii* have both experienced a recent Whole Genome Duplication (WGD), a role of that event in the origination of the two shorter species-specific genes is plausible. The undetectable homologue in *T. phaphii*, another post-WGD species, has both similar CDS length to that of the pre-WGD ones and conserved sequence similarity to them, which is consistent with a link between shortening and loss of sequence similarity.

Finally, we investigated how orphan genes that have originated by divergence beyond recognition might impact human biology. Our approach identified thirteen human genes that underwent complete divergence along the human lineage and have no detectable homologues outside of mammals (see *Figure 8—source data 1*). Examining the ENSEMBL and UniProt resources revealed that three of these thirteen genes are associated with known phenotypes. One of them is ATP-synthase membrane subunit 8 (*MT-ATP8*), which has been implicated with infantile cardiomyopathy (*Ware et al., 2009*) and Kearns-Sayre syndrome (*van der Westhuizen et al., 2010*) among other diseases. The other two are both associated with cancer: *DEC1* (*Nishiwaki et al., 2000*) and *DIRC1* (*Druck et al., 2001*). It is curious that three out of three of these genes are associated with disease, two of which with cancer, although the small number prevents us from drawing conclusions. Nonetheless, this observation is consistent with previous findings showing that multiple novel human genes are associated with cancer and cancer outcomes suggesting a role for antagonistic evolution in the origin of new genes (*McLysaght and Hurst, 2016*).

## Discussion

The persistent presence of orphans and TRGs in almost every genome studied to date despite the growing number of available sequence databases demands an explanation. Studies in the past 20 years have mainly pointed to two mechanisms: de novo gene emergence and sequence divergence of a pre-existing gene, either an ancestrally present or one acquired by horizontal transfer. However, the relative contributions of these mechanisms have remained elusive until now. Here, we have specifically addressed this problem and demonstrated that sequence divergence of ancestral genes explains only a minority of orphans and TRGs.

We were very conservative when estimating the proportion of orphans and TRGs that have evolved by complete divergence inside regions of conserved micro-synteny. Indeed, we simultaneously underestimated the number of orphans and TRGs while overestimating the number that

originated by divergence. We underestimated the total number of orphans and TRGs by relying on relaxed similarity search parameters. As a result, we can be confident that those genes without detectable similarity really are orphans and TRGs, but in turn we also know that some will have spurious similarity hits giving the illusion that they have homologues when they do not in reality. Furthermore, the annotation that we used in yeast does not include the vast majority of dubious ORFs, labelled as such because they are not evolutionarily conserved even though most are supported by experimental evidence (*Li et al., 2008*).

We overestimated the number of genes that have undergone complete divergence by assuming that all genes in conserved micro-synteny regions share common ancestry. There are however limitations in using synteny to approximate common descent. First, with time, genome rearrangements shuffle genes around and synteny is lost. This means that when comparing distantly related species, the synteny signal will be more tenuous and eventually completely lost. Second, combinations of evolutionary events can place non-homologous genes in directly syntenic positions. Loss of a gene in a lineage followed by tandem duplication of a neighbouring gene, translocation of a distant one, or de novo emergence, could potentially contribute to placing in syntenic positions pairs of genes that are not in fact homologous. As such, the results of our pipeline can be viewed as an upper bound estimate of the true rate of divergence beyond recognition.

Previous efforts to measure the rate of complete divergence beyond recognition have done so using simulations (*Vakirlis et al., 2018*; *Moyers and Zhang, 2014*; *Albà and Castresana, 2007*; *Moyers and Zhang, 2016*; *Jain et al., 2019*), within a different context and with different goals, mainly to measure 'BLAST error'. Interestingly, our estimates are of the same order of magnitude as previous results from simulations (*Moyers and Zhang, 2014*; *Moyers and Zhang, 2016*). Nonetheless, using the term 'BLAST error' or talking about 'false negatives' would be epistemologically incorrect in our case. When focusing on the outcome of divergence itself, it is clear that once all sequence similarity has been erased by divergence, BLAST, a *similarity* search tool, should not be expected to detect any.

Simulation-based studies have been valuable in quantifying the link between evolutionary distance and absence of sequence similarity. They are however limited in that they can only show that sequence divergence *could* explain a certain proportion of orphans and TRGs, not that it actually *does* explain it. Making the jump from 'could' to 'does' requires the assumption that divergence beyond recognition is much more plausible than, for example, de novo emergence. This is a prior probability which, currently, is at best uncertain. Our approach, on the other hand, does not make assumptions with respect to the evolutionary mechanisms at play, that is we do not need prior knowledge of the prevalence of divergence beyond recognition to obtain an estimate.

Many studies have previously reported that genes without detectable homologues tended to be shorter than conserved ones (*Tautz and Domazet-Lošo, 2011*; *Wissler et al., 2013*; *Toll-Riera et al., 2009*; *Ekman and Elofsson, 2010*; *Palmieri et al., 2014*; *Vakirlis et al., 2016*; *Khalturin et al., 2009*). This relationship has been interpreted as evidence that young genes can arise de novo from short open reading frames (*McLysaght and Guerzoni, 2015*; *Carvunis et al., 2012*; *Zhao et al., 2014*; *Siepel, 2009*) but also as the result of a bias due to short genes having higher evolutionary rates, which may explain why their homologues are hard to find (*Moyers and Zhang, 2014*; *Moyers and Zhang, 2017*). Our results enable another view of these correlations of evolutionary rate, gene age and gene length (*Tautz and Domazet-Lošo, 2011*; *Wolf et al., 2009*; *Albà and Castresana, 2005*). We have shown that an event akin to incomplete pseudogenization could be taking place, wherein a gene loses functionality through some disruption, thus triggering rapid divergence due to absence of constraint. After a period of evolutionary 'free fall' (*Wolf et al., 2009*), this would eventually lead to an entirely novel sequence. If this is correct, then it could explain why some short genes, presenting as young, evolve faster.

Disentangling complete divergence from other processes of orphan and TRG origination is non-trivial and requires laborious manual inspection (*Prabh and Rödelsperger, 2019*; *Zhou et al., 2008*). Our approach allowed us to explicitly show that divergence can produce homologous genes that lack detectable similarity and to estimate the rate at which this takes place. We are able to isolate and examine these genes when they are found in conserved micro-synteny regions, but at this point we have only a statistical global view of the process of divergence outside of these regions. Since, for example, in yeast and in Arabidopsis,~50% of orphan genes are located outside of syntenic regions of near relatives (*Arendsee et al., 2019*), the study of their evolutionary origins

represents exciting challenges for future work. Why do genes in yeast and fly appear to reach the 'twilight zone' of sequence similarity considerably faster than human? One potential explanation is an effect of generation time and/or population size on evolutionary rates (*Martin and Palumbi, 1993*; *Bromham and Penny, 2003*) and thereby on the process of complete divergence.

Overall, our findings are consistent with the view that multiple evolutionary processes are responsible for the existence of orphan genes and suggest that, contrary to what has been assumed, divergence is not the predominant one. Investigating the structure, molecular role, and phenotypes of homologues in the 'twilight zone' will be crucial to understand how changes in sequence and structure produce evolutionary novelty.

## Materials and methods

All data and scripts necessary to reproduce all figures and analyses are available at https://github.com/Nikos22/Vakirlis_Carvunis_McLysaght_2019 (*Vakirlis, 2020*; copy archived at https://github.com/elifesciences-publications/Vakirlis_Carvunis_McLysaght_2019). Correspondence of scripts to figures can be found in each Materials and methods subsection and in the readme file available online on GitHub.

### Data collection

Reference genome assemblies, annotation files, CDS and protein sequences were downloaded from NCBI's GenBank for the fly and yeast datasets, and ENSEMBL for the human dataset. Species names and abbreviations used can be found in *Table 1*. The latest genome versions available in January 2018 were used. The yeast annotation used did not include dubious ORFs. OrthoDB v 9.1 flat files were downloaded from https://www.orthodb.org/?page=filelist. Divergence times for focal-target pairs were obtained from http://timetree.org/ (*Hedges et al., 2006*) (estimated times). $d_N$ and $d_S$ values where obtained for *D. melanogaster* and *D. simulans* from http://www.flydivas.info/ (*Stanley and Kulathinal, 2016*) and for human and mouse from ENSEMBL biomart. For *S. cerevisiae*, we calculated $d_N$ and $d_S$ over orthologous alignments of 5 *Saccharomyces* species (*S. cerevisiae, S. paradoxus, S. mikatae, S. kudriavzevii, S. bayanus*) downloaded from http://www.saccharomycessensustricto.org/cgi-bin/s3.cgi (*Scannell et al., 2011*) using *yn00* from PAML (*Yang, 2007*) (average of 4 pairwise values for each gene).

### Synteny-based pipeline for detection of homologous gene pairs

1. *Data preparation:* Initially, OrthoDB groups were parsed and those that contained protein-coding genes from the focal species were retained. OrthoDB constructs a hierarchy of orthologous groups at different phylogenetic levels, and so we selected the highest one to ensure that all relevant species were included. For every protein-coding gene in the annotation GFF file of the three focal species (yeast, fly, human), we first matched its name to its OrthoDB identifier. Then, we stored a list of all the target species genes found in the same OrthoDB group for every focal gene. Finally, the OrthoDB IDs of the target genes too were matched to the annotation gene names.
2. *BLAST similarity searches*: All similarity searches were performed using the BLAST+ (*Altschul et al., 1997*) suite of programs. Focal proteomes were used as query to search for similar sequences, using BLASTp, against their respective target proteomes. The search was performed separately for every focal-target pair. Default parameters were used and the *E-value* parameter was set at 1. Target proteomes were reversed using a Python script and the searches were repeated using the reversed sequences as targets. The results from the reverse searches were used to define 'false homologies'.
3. *Identification of regions of conserved micro-synteny:* For every focal-target genome pair, we performed the following: for every chromosome/scaffold/contig of the focal genome, we examine each focal gene in a serial manner (starting from one end of the chromosome and moving towards the other). For each focal protein-coding gene, if it does not overlap more than 80% with either its +1 or −1 neighbour, we retrieve the homologues of its +1,+2 and −1,−2 neighbours in the target genome, from the list established previously with OrthoDB (*Kriventseva et al., 2008*). We then examine every pair-wise combination of the +1,+2 and −1,−2 homologues and identify cases were the +1,−1 homologues are on the same chromosome and are separated by either one or two protein-coding genes. Out of these candidates,

we only keep those for which, if it exists, the homologue of the −2 neighbour is adjacent or separated by one gene from the homologue of the −1 neighbour, and the homologue of the +two neighbour, if it exists, is adjacent or separated by one gene from the homologue of the +one neighbour. We further filter out all cases for which the homologues of +1 and −1 belong in the same OrthoDB group, that is they appear to be paralogues. The intervening gene(s) 'opposite' the focal gene (between the homologues of its −1 and +one neighbours) are stored in a list. The specific choice of synteny criterion was made after we conducted an initial trial with a minimum of one homologue on either side, which showed limited false positives, revealed by visual inspection (obvious cases of non-homologous genes which, due to rearrangements such as micro-inversions were placed 'opposite' each other, even though they did not originate by divergence). One of these false positives was the well-studied de novo yeast gene *BSC4* (*Cai et al., 2008*). Requiring the additional syntenic homologous gene on both sides (when the homologues exist), provided a balanced solution: it removed the *BSC4* gene along with all other identified similar cases, again verified by extensive visual inspection and comparison to other genomic synteny resources (ENSEMBL, Yeast Gene Order Browser), while retaining a number of conserved micro-syntenic regions that was high enough to allow us to perform our analyses. To explore the impact of the synteny definition on our main finding, namely that divergence accounts for a minority of orphan genes, we replicated the analysis using one more relaxed and one more stringent synteny criterion. The relaxed definition considered only one syntenic homologue on either side, with no examination of the homologues of the −2,+2 neighbors. The stringent definition added an additional syntenic homologue to either side relative to the current one: three genes on either side need to have syntenic homologues that are separated by one gene at most and, again, we retain cases where the outer neighbors (−3,+3) have no homologues at all to the target genome.

4. *Identification of similarity:* Once all the focal genes for which a region of conserved micro-synteny has been identified have been collected for a focal-target genome pair, we test whether similarity can be detected at a given E-value threshold. First, we look at whether a precomputed (previously, by us, whole proteome-proteome comparison) BLASTp match exists between the translated focal gene and the its translated 'opposite' genes (taking into account all translated isoforms), where we predict the match should be found most of the time. If no match exists at the amino acid level there, we perform a TBLASTN search with default parameters, using the focal gene as query and the genomic region of the 'opposite' gene plus the 2 kb flanking regions as target. The search is repeated using the reversed genomic region as target. If no match is found, we look whether a BLASTP match exists to any translated gene of the target genome. Finally, for the genes for which no similarity has been detected, we perform a TBLASTN search against the entire genome of the target species. This final TBLASTN step is not included in the setting of the optimal E-value and a fixed E-value threshold of $10^{-6}$ is used.

Related to *Figure 3*, *Figure 4*, *Figure 2—figure supplement 1*, *Figure 6—figure supplement 2*; relevant scripts: Figure3A.R, Figure3B_4_fig2-supp1.R.

## Calculation of undetectable and false homologies and definition of optimal E-values

For every focal-target pair and for every E-value cut-off, the proportions of focal genes with at least one identified region of conserved micro-synteny for which a match was found 'opposite' or elsewhere in the genome were calculated. The remaining proportion, that is those with conserved micro-synteny but no match, constitutes the percentage of putative undetectable homologies. To estimate the 'false homologies', we calculated the proportion of the focal proteome that had a BLASTp match to the reversed target proteome, or to their corresponding reversed syntenic genomic region for the ones with identified micro-synteny (see step 4 of previous section). Based on these proportions, we chose the highest value limiting 'false homologies' to 0.05 for our analyses.

We also calculated the Mathews Correlation Coefficient (MCC) measure of binary classification accuracy for every E-value cut-off. This is a balanced measure that takes into account true and false positives and negatives which can be used even in cases of extensive class imbalance. At every E-value cut-off, we treated undetectable homologies as False Negatives, and false homologies (matches to the reversed proteome) as False Positives. Similarly, sequence-detected homologies (defined based on micro-synteny) were treated as True Positives and genes for which the reversed-search gave no significant hit were treated as True Negatives. The MCC measure was then

calculated at each E-value cut-off based on these four values using the *mcc* function of R package mltools. When multiple E-value cut-offs had the same MCC (rounded at the 3$^{rd}$ decimal), the highest (less stringent) E-value was retained. The results for each focal-target genome pair are shown in *Figure 3—source data 1* ('general E-value' column).

Related to *Figure 3*, *Figure 4*; relevant scripts: Figure3A.R, Figure3B_4_fig2-supp1.R, Balanced_optimal_evalue_MCC.R.

## Calculation of contribution of divergence beyond recognition to observed numbers of genes without detectable similarity

For a given pair of focal-target genomes, we estimate the proportion of all focal genes without detectable similarity that is due to processes other than sequence divergence in a pairwise manner (*Figure 6*) and in a phylogeny-based manner (*Figure 6—figure supplement 3*). The pairwise approach is calculated as follows (see also *Figure 6—figure supplement 1* for a schematic explanation): an $X$ number of the total $n$ of focal genes will have no similarity with the target, based on a BLASTP search of the target's proteome using the corresponding optimal E-value cut-off and a TBLASTN search of the target's genome with an E-value cut-off of $10^{-6}$. We have also estimated the proportion $d$ of total genes that have lost similarity due to divergence. This was calculated over genes in conserved micro-synteny but we assume that it can be used as a proxy for the entire genome since presence in a conserved micro-syntenic region does not significantly impact evolutionary rates (*Figure 5*). By calculating the ratio of $d$ over $X/n$ we can obtain the contribution of divergence to the total genes without similarity. Note that, for this calculation to be unbiased, $d$ here is based on the same similarity searches used to define $X$ (i.e. it does not include the local TBLASTN search shown in step 4 of *Figure 2B*). The phylogeny-based approach is performed as follows: for a given 'phylostratum' (a given ancestral branch of the focal species), we estimate the proportion of genes restricted to this phylostratum due to divergence, again calculated over genes in conserved micro-synteny and extrapolated to all genes as in the pairwise case. This is done by taking the number of genes restricted to the phylostratum (TRGs, i.e. those for which the phylogenetically farthest species with a sequence similarity match falls within the subtree defined by the phylostratum) that have a putative undetectable homologue (based on micro-synteny) in at least one lineage outside of that phylostratum, and dividing them by the number of all genes that are predicted to have a homologue (based on micro-synteny) in at least one lineage outside the phylostratum. In other words, the proportion out of all genes with at least one micro-synteny conserved region, and thus a putative homologue, with a species outside the phylostratum, that are restricted, based on sequence similarity, within the specific phylostratum. As in the pairwise case, this proportion is compared to the proportion calculated based on sequence similarity alone out of all genes, meaning the proportion of TRGs for a given phylostratum, out of all genes.

The proportion of TRGs that we predict can be explained by divergence at the phylostrata of Saccharomyces (*S. kudriavzevii*, *S. arboricola*), melanogaster subgroup (*D. simulans*, *D. sechellia*, *D. yakuba*, *D. erecta*, *D. ananassae*) and primates (*P. trogrolydes*, *G. gorilla*) is obtained by the phylogeny-based approach described above, at the phylostrata with branches of origin at 15, 37 and 9 million years ago respectively.

Related to *Figure 5*, *Figure 6*, *Figure 6—figure supplements 1–3*; relevant scripts: Figure6_fig6-supp2.R, Figure6-supp3.R, Figure_5_7_8.R.

## Protein and CDS properties

Pfam matches were predicted using *PfamScan.pl* to search protein sequences against a local Pfam-A database downloaded from ftp://ftp.ebi.ac.uk/pub/databases/Pfam (*Finn et al., 2016*; *Eddy, 2011*). Guanine Cytosine content and CDS length was calculated from the downloaded CDSomes in Python. Secondary structure (Helix, Strand, Coil), solvent accessibility (buried, exposed) and intrinsic disorder were predicted using *RaptorX Property* (*Wang et al., 2016*). Transmembrane domains were predicted with *Phobius* (*Käll et al., 2007*). Low complexity regions in protein sequences were predicted with *segmasker* from the BLAST+ suite. In the correlation analysis of the various properties, when multiple isoforms existed for the focal or target gene in a pair, we only kept the pairwise combination (focal-target) with the smallest CDS length difference. For the protein and CDS properties analyses, we removed 23 pairs of undetectable homologues for which our bioinformatic pipeline

failed to retrieve both the focal and target species homologue CDS sequence due to non-correspondence between the downloaded annotation and CDS files. Furthermore, in all undetectable homologues property analyses, we removed from our dataset 14 pairs of undetectable homologues whose proteins consisted of low complexity regions in more than 50% of their length, since we observed that such cases can often produce false positives (artificial missed homologies) because of BLASTP's low complexity filter. Pairwise alignments were performed with MAFFT (*Katoh and Standley, 2013*). All statistical analyses were conducted in R version 3.2.3. All statistical tests performed are two-sided.

Related to *Figure 7*; relevant scripts: Figure_5_7_8 .R.

### Identification of TRGs resulting from lineage-specific divergence within micro-syntenic regions

To identify novel genes likely resulting from lineage-specific divergence and restricted to a specific taxonomic group, we applied the following criteria. Out of all the candidate genes in the three focal species with at least two undetectable homologues in two non-monophyletic (non-sister) target species, we retained those that had no match, according to our pipeline, to target species that diverged before the most distant of the target species with an undetectable homologue (see *Figure 8A* for a schematic representation). For those genes, we also performed an additional BLASTP search against NCBI's NR database with an E-value cut-off of 0.001 and excluded genes that had matches in outgroup species (i.e. in species outside of Saccharomyces, *Drosophila* and placental mammals for yeast, fly and human respectively).

Related to *Figure 8*; relevant scripts: Figure_5_7_8 .R.

## Acknowledgements

The authors are grateful to Drs. Gilles Fisher, Ingrid Lafontaine, Laurence Hurst and Aaron Wacholder for reading the manuscript prior to submission.

## Additional information

### Funding

| Funder | Grant reference number | Author |
|---|---|---|
| Seventh Framework Programme | 309834 | Aoife McLysaght |
| Seventh Framework Programme | 771419 | Aoife McLysaght |
| National Institute of General Medical Sciences | R00GM108865 | Anne-Ruxandra Carvunis |
| Kinship Foundation | Searle Scholars Program | Anne-Ruxandra Carvunis |

The funders had no role in study design, data collection and interpretation, or the decision to submit the work for publication.

### Author contributions

Nikolaos Vakirlis, Conceptualization, Data curation, Software, Formal analysis, Investigation, Methodology; Anne-Ruxandra Carvunis, Aoife McLysaght, Conceptualization, Supervision, Funding acquisition, Methodology

### Author ORCIDs

Nikolaos Vakirlis https://orcid.org/0000-0001-7606-6987
Anne-Ruxandra Carvunis https://orcid.org/0000-0002-6474-6413
Aoife McLysaght https://orcid.org/0000-0003-2552-6220

**Decision letter and Author response**
Decision letter https://doi.org/10.7554/eLife.53500.sa1
Author response https://doi.org/10.7554/eLife.53500.sa2

## Additional files

### Supplementary files
• Transparent reporting form

### Data availability
Data is available in the Supplementary Information and in Source Data files. All data and scripts necessary to reproduce all figures and analyses are also available at https://github.com/Nikos22/Vakirlis_Carvunis_McLysaght_2019 (copy archived at https://github.com/elifesciences-publications/Vakirlis_Carvunis_McLysaght_2019). Correspondence of scripts to figures can be found in each Methods subsection and in the readme file available online on GitHub.

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
