## [Decision Letter]

**Acceptance summary:**

When the first orphan genes were discovered in the yeast genome project, there was a discussion that suggested that genes that do not match with other genes in the database would become fewer over time, once the databases are better filled. Now the databases are filled, but orphan genes are still identified in every evolutionary lineage. Hence, the discussion turned into the assumption that this is due to a lack of sensitivity of detecting remote homologues and that better algorithms need to be devised. Vakirlis et al. now argue that a careful analysis of syntenic genome stretches allows to distinguish between fast sequence divergence beyond recognition and de novo evolution out of non-coding stretches of DNA. By running this comparison in three major evolutionary lineages, they show that only about a third of the genes may be orphans due to rapid divergence, while the remainder are likely truly newly evolved orphan genes.

**Decision letter after peer review:**

[Editors’ note: the authors submitted for reconsideration following the decision after peer review. What follows is the decision letter after the first round of review.]

Thank you for submitting your work entitled "Synteny-based analyses indicate that sequence divergence is not the dominant source of orphan genes" for consideration by *eLife*. Your article has been reviewed by three peer reviewers, and the evaluation has been overseen by a Reviewing Editor and a Senior Editor. The following individuals involved in review of your submission have agreed to reveal their identity: Eve Syrkin Wurtele (Reviewer #3).

Our decision has been reached after consultation between the reviewers. Unfortunately, we have to reject the paper at this stage, since the submission did not include the results tables that were used to generate the figures and conclusions. Please note that all primary results that are required to reproduce your conclusion need to be accessible and properly listed in the manuscript.

Apart of this, the reviewers were mostly positive, although some critical issues need close attention. In addition to the points raised by the reviewers, the following points need to be addressed for a possible re-submission (please provide line numbers in you next submission):

Taxon choice:

The *Drosophila* set includes *C. elegans* as an outgroup, which is not very appropriate. Although both belong to Ecdysozoa, the actual divergence of *C. elegans* to humans is probably less than to *Drosophila*, since evolutionary rates in the vertebrate lineage are much smaller than in the insect lineage (as it is also shown in the present analysis). Hence, *C. elegans* comparisons need to be removed.

The part on "Selecting the optimal E-value cutoff" is not really novel and has also no new results. It should be moved to supplementary material (with the updates suggested by Tomi).

"(we find no, or very limited, difference in evolutionary rate between the two groups in terms of dN, dS, dN/dS; see Materials and methods and Supplementary figure 4)." – this is a key statement that needs to be properly presented in the text, including appropriate statistics. To make it valid, it would have to consider the possibility of different rates in different pairwise comparisons. (See also the comments by reviewer #2)

"We therefore searched ENSEMBL and UniProt for phenotypes and involvement in

disease for the ten genes within micro-synteny regions that we predict originated through complete divergence along the human lineage" – this is a standard orphan gene search, unclear what it adds to the message in the present paper.

"We find that at most 33% of orphans and TRGs have possibly originated by complete

divergence." – it is not completely clear where this number comes from.. It needs to be properly justified, given that it is the key message of the paper.

Reviewer #1:

This is important work that tries to estimate relative contribution of sequence divergence in the origin of novel genes (also known as orphan genes or taxonomically restricted genes). To my knowledge this is the first study entirely focused on this question which is essential for understanding which mechanism (divergence or de novo route) prevails in creating evolutionary innovations at the genome/proteome level. The authors devised a very neat protocol that detects conserved synteny regions between species and then looks for proteins that lack sequence similarity in the target genome but are otherwise embedded within these conserved synteny regions. By extrapolating the findings from the conserved synteny regions to the whole genome the authors conclude that divergence mechanism accounts for at most a third of eukaryotic novel genes.

1) Subsection “Selecting optimal BLAST E-value cut-offs”, Figure 3A – It is really elegant how the authors estimated the proportion of false positive and false negative homologues. From their figures it is clear that the proportion of false positives changes quite differently from the proportion of false negatives depending on the e-value. The authors mention this in the text, but they don't discuss that controlling of false positives is more critical when deciding on the appropriate e-value cutoffs. This is evident from the shape of the Figure 3A curves, where false negatives show linear-like dependence and false positives exponential-like dependence, plus the false positive curves are not dependent on the evolutionary distance. In practice, this means that failing to properly control for false positives would generate spurious sequence similarity hits all over the place, whereas not perfectly adjusted e-value would not have a such profound effect on false negatives, especially in the phylogenetic context.

2) Linked to the previous comment: The authors discuss the BLAST "false negatives" debate but miss the balance by omitting to cite Domazet-Loso et al., 2017 paper which stresses the importance of evaluating BLAST "false positives". Given that the manuscript specifically deals with the false positives this paper should be cited.

3) Subsection “Calculation of undetectable and false homologies and definition of optimal E-values” – I had problems to understand how Mathews Correlation Coefficient was used to decide on the E-value cut-off. Could you please describe the protocol in more details here?

4) Subsection “The rate of “divergence beyond recognition” and its contribution to the

total pool of genes without similarity” – Presentation of the results in Figure 4B and 4C (Figure 6—figure supplement 2) is quite confusing.

5) Subsection “Calculation of proportion of orphan genes due to processes other than sequence divergence” – "This is done by taking…" This part were phylogeny-based proportions are obtained was not comprehensible to me. Could you please make some schematic representation of this part of the protocol?

Reviewer #2:

The authors have made a comprehensive attempt to quantify the contribution of sequence divergence as a source of orphan genes. They have ingeniously used micro-synteny to identify the orphan genes created by divergence. They estimate that at most one-third of the genes within the micro-syntenic regions result from sequence divergence. Subsequently, they extrapolate this result to the entire genome and conclude that the majority of orphan genes are not formed through sequence divergence. Although several studies have investigated orphan genes in various organisms, the relative contribution of processes such as de novo gene creation and sequence divergence in the formation of orphan genes remains largely unexplored. Hence, I consider that this work can significantly contribute to the progress of the field.

Major concern:

The authors assume that the similar distribution of evolutionary rates within and outside the syntenic blocks indicates that the proportion of orphan genes created through sequence divergence within the syntenic block can be extrapolated to the whole genome (Subsection “The rate of “divergence beyond recognition” and its contribution to the total pool of genes without similarity” paragraph 2). Given that the evolutionary rates cannot be estimated for the orphan genes lacking detectable homologs, the assumption made by the authors about the comparability of evolutionary rates across the genome is most likely based on a dataset that excludes such genes. Thus, their argument, although correct for the overall distribution of evolutionary rates, should not be used to extrapolate the proportion of orphan genes originated by divergence within the syntenic blocks to the whole genome.

Reviewer #3:

Orphan genes are an exciting addition to our understanding of speciation and adaptation of organisms to new environments. This research represents a unique and important approach to validate that rapidly-changing genes exist. The manuscript is well-written and clearly presents a very complex set of concepts.

The authors did something new. Rather than assuming (as many have done, but with no basis for this assumption) that any orphan gene that cannot positively be IDed as de novo is "rapidly changing divergent duplicates" the research in this manuscript take the opposite tact, and directly IDs the orphans genes that have arisen by rapid evolution. Indeed, this manuscript is the best evidence for orphans that arose via a " rapidly changing divergent duplicate gene mechanism".

Providing positive evidence for the set of genes that have rapidly diverged is particularly important because it opens the path for researchers to explore the mechanisms whereby particular genes can evolve so much more quickly than the typical gene.

The table with the abbreviations (e.g., ggor) should be moved from supplementary to the main text. That way the reader doesn't need to access supplementary to understand Figure 6.

In both yeast and Arabidopsis, ~50% of orphan genes are NOT located in (micro) syntenic regions of near relatives (Arendsee et al., 2019). These genes (and how they arose) are really interesting as well. The authors might want to mentioned this in the Discussion.

The manuscript mentions genes with "retention of structural similarity " that "suggest the possibility of conservation of ancestral signals in the absence of sequence similarity." Keeping at least some of a structure but losing homology!! Cool. How might this fit into evolution? or is it just a quirk?

The vocabulary the authors use (e.g., "twilight zone", "freefall") adds to the paper.

[Editors’ note: further revisions were suggested prior to acceptance, as described below.]

Thank you for submitting your article "Synteny-based analyses indicate that sequence divergence is not the main source of orphan genes" for consideration by *eLife*. Your article has been reviewed by three peer reviewers, and the evaluation has been overseen by Diethard Tautz as the Senior Editor. The following individuals involved in review of your submission have agreed to reveal their identity: Neel Prabh (Reviewer #2); Eve Syrkin Wurtele (Reviewer #3).

The reviewers have discussed the reviews with one another and have come to the conclusion that the manuscript is basically acceptable. Only one reviewer has some small concerns and this review is provided in full below. Given that you had already indicated in your response letter that this issue is minor, we would hope that you can easily integrate this into your manuscript and that no further reviewing will be necessary.

Reviewer #2:

The authors have made a sincere effort to highlight the caveats of their method and adjusted their manuscript as requested by the reviewers. The only remaining concern I have is with the formula used to calculate the contribution of divergence (Figure 6—figure supplement 1). It’s a ratio of two ratios. The numerator is the ratio of orphan genes to all genes within the syntenic blocks, and the denominator is the ratio of orphan genes to all genes across the genome. Here, the criterion to define a syntenic block is pivotal, and to ensure that false positives are excluded the authors limit the block to minimum two homologues on either side of the focal gene separated by either one or two genes. This criterion leaves the majority of genes outside the syntenic blocks in all pairwise comparisons, especially the "match not found" genes (Figure 2—figure supplement 1, Figure 3—source data 3: Figure 3—figure supplement 1).

The authors accept that the proportion of orphan genes within the syntenic blocks decreases as the criterion is made more stringent. They suggest this is because a lesser number of genes were found within the syntenic regions, but the indicated change in the proportion of orphan genes suggests that as more genes are included within syntenic blocks, the proportion of orphan genes within the blocks rises. This is quite intuitive because in many pairwise comparisons (19/48) the number of "match not found" within syntenic blocks is in single-digit and can potentially increase many folds by the mere addition of few more genes (Supplementary Table: Figure 3—figure supplement 1).

Thus, with more relaxed criteria, the calculated divergence contribution will increase as higher fractions of orphan genes are included within syntenic blocks, but the total fraction of orphan genes does not change. Here, the compatibility of the evolutionary rates within and outside the blocks can still be maintained, while the calculated contribution made by divergence will vary.

In the rebuttal, the authors write:

"Note that, although, as expected, stricter synteny criteria led to fewer genes being found in conserved micro-syntenic blocks, overall results changed minimally between the two versions and hence can be considered robust."

It has to be clearly shown, that increasing the stringency of the synteny criterion does not specifically exclude "match not found" genes and their proportion within the syntenic blocks is not a function of this criterion. The authors are advised to make this clarification within the manuscript before making genome level extrapolation. The method established by the authors is robust, and certainly, it will be extensively used in the future. However, the formula used for the genome-level extrapolation appears extremely sensitive to the synteny criterion, unless shown otherwise. Given that the authors have data supporting that synteny criterion does not affect the overall result, they should show it.

---

## [Author Response]

[Editors’ note: what follows is the authors’ response to the first round of review.]

Apart of this, the reviewers were mostly positive, although some critical issues need close attention. In addition to the points raised by the reviewers, the following points need to be addressed for a possible re-submission (please provide line numbers in you next submission):Taxon choice:The Drosophila set includes *C. elegans* as an outgroup, which is not very appropriate. Although both belong to Ecdysozoa, the actual divergence of *C. elegans* to humans is probably less than to Drosophila, since evolutionary rates in the vertebrate lineage are much smaller than in the insect lineage (as it is also shown in the present analysis). Hence, *C. elegans* comparisons need to be removed.

We thank the reviewers for this comment. We have now removed *C. elegans* and all comparisons involving it from the article.

The part on "Selecting the optimal E-value cutoff" is not really novel and has also no new results. It should be moved to supplementary material (with the updates suggested by Tomi).

While we understand the reason for this comment (the same idea has indeed been explored in the past) we feel that this approach is novel in methodology and in phylogenetic space. Additionally, reviewer #1 found this part interesting and “elegant” and requested that we discuss the impact of false positives compared to false negatives some more. To balance these different viewpoints, we have significantly shortened the relevant section of the manuscript and made the novelty more evident. We note that it is particularly interesting that, with our updated methodology and expanded species comparisons, we arrive at similar conclusions as past efforts, including those of this reviewer.

Here is the updated text from the revised manuscript:

“Homology detection is highly sensitive to the technical choices made during sequence similarity searches7,27. We therefore sought to explore how the choice of E-value threshold would impact interpretations of divergence beyond similarity. First, we performed BLASTP searches of the focal species’ total protein sequences against the total reversed protein sequences of each target species. Matches produced in these searches can safely be considered “false homologies” since biological sequences do not evolve by reversal28 (see Materials and methods). These false homologies were then compared to “undetectable homologies”: cases with conserved micro-synteny (presumed homologues) but without any detectable sequence similarity. In Figure 3A, we can see how the ratios of undetectable and false homologies vary as a function of the BLAST E-value threshold used. The proportion of undetectable homologies depended quasi-linearly on the E-value cut-off. By contrast, false homologies depended exponentially on the cut-off, as expected from the E-value definition. Furthermore, the impact of E-value cut-off was more pronounced in comparisons of species separated by longer evolutionary distances, whereas it was almost non-existent for comparisons amongst the most closely related species. Conversely, there seems to be no dependence between percentage of false homologies and evolutionary time across the range of E-values that we have tested (all lines overlap in the graphs in the bottom panel of Figure 3A). This means that, when comparing relatively closely related species, failing to appropriately control for false homologies would have an overall more severe effect on homology detection than failing to account for false negatives. In the context of phylostratigraphy (estimation of phylogenetic branch of origin of a gene based on its taxonomic distribution29), gene age underestimation due to BLAST “false negatives” has been considered a serious issue30, although the importance of spurious BLAST hits generating false positives has also been stressed31. We defined a set of E-value cut-offs optimised for phylostratigraphy, by choosing the highest E-value that keeps false homologies under 5%. This strategy emphasizes sensitivity over specificity. We have also calculated general-use optimal E-values by using a balanced binary classification measure (see Materials and methods). The phylostratigraphy optimal E-value thresholds are 0.01 for all comparisons using yeast and fly as focal species and 0.001 for those of human, except for chimpanzee (10-4). These are close to previously estimated optimal E-value cut-offs for identifying orphan genes in *Drosophila*, found in the range of 10-3 – 10-5, see ref 32. These cut-offs have been used for all downstream analyses. We find that, for the vast majority of focal genes examined that do have matches, the match occurs in the predicted region (“opposite”), i.e., within the region of conserved micro-synteny. In 36/48 pair-wise species comparisons, at least 90% of the focal genes in micro-synteny for which at least one match was eventually found in the target genome, a match was within the predicted micro-syntenic region (Figure 3B). This finding supports the soundness of our synteny-based approach for homologue identification.”

"(we find no, or very limited, difference in evolutionary rate between the two groups in terms of dN, dS, dN/dS; see Materials and methods and Supp Figure 4)." – this is a key statement that needs to be properly presented in the text, including appropriate statistics. To make it valid, it would have to consider the possibility of different rates in different pairwise comparisons. (See also the comments by reviewer #2)

We thank the reviewers for this important remark. Detailed statistics including effect sizes and better presentation of the results of evolutionary rate comparisons have now been added to the main text of the revised manuscript (Figure 5). We have added the following text:

“We calculated the non-synonymous (dN) and synonymous (dS) substitution rates of genes found inside and outside regions of conserved micro-synteny relative to closely related species (Materials and methods). […] Despite these minimal caveats, evolutionary rates are globally very similar inside and outside regions of conserved micro-synteny, allowing to extrapolate with confidence.”

We were not entirely sure what the reviewers mean in the clause “consider the possibility of different rates in different pairwise comparisons.”. We interpreted it as a concern over evolutionary rates varying across the different focal-target pairwise species comparisons and therefore not being captured by our dN and dS calculations. If this is an accurate interpretation of the comment, we would like to reassure the reviewer that, even if such a difference existed, it would not affect the comparison of rates of genes inside and outside of conserved syntenic blocks.

"We therefore searched ENSEMBL and UniProt for phenotypes and involvement indisease for the ten genes within micro-synteny regions that we predict originated through complete divergence along the human lineage" – this is a standard orphan gene search, unclear what it adds to the message in the present paper.

We thank the reviewers for this comment. The way this was written in our original submitted manuscript did not make sufficiently clear the motivation and purpose of the search. The point of this analysis is not a general scan of UniProt and ENSEMBL, but rather a search for how the specific taxonomically restricted genes that have arisen in the human lineage through complete sequence divergence impact human biology, and especially evidence of involvement in disease. Since it has been reported that human novel genes in general are often associated to cancer and cancer outcomes, it is worth seeing whether our results seem to support previous findings or not. Since our study is the first to isolate novel genes that have evolved by complete divergence, this is the first look at the function of these genes in the human lineage. We have now rephrased the relevant part of the manuscript to clarify this point:

“Finally, we investigated how orphan genes that have originated by divergence beyond recognition might impact human biology. Our approach identified thirteen human genes that underwent complete divergence along the human lineage and have no detectable homologues outside of mammals (see Figure 8—figure supplement 1). Examining the ENSEMBL and UniProt resources revealed that three of these thirteen genes are associated with known phenotypes. One of them is ATP-synthase membrane subunit 8 (MT-ATP8), which has been implicated with infantile cardiomyopathy40 and Kearns-Sayre syndrome41 among other diseases. The other two are both associated with cancer: DEC142 and DIRC143. It is curious that three out of three of these genes are associated with disease, two of which with cancer, although the small number prevents us from drawing conclusions. Nonetheless, this observation is consistent with previous findings showing that multiple novel human genes are associated with cancer and cancer outcomes suggesting a role for antagonistic evolution in the origin of new genes44.”

"We find that at most 33% of orphans and TRGs have possibly originated by completedivergence." – it is not completely clear where this number comes from. It needs to be properly justified, given that it is the key message of the paper.

We thank the reviewers for pointing out this insufficiency. This figure (which in the revised manuscript has dropped to 30% due to removal of *C. elegans*) is the average proportion, across the three datasets, of TRGs originated through divergence as estimated by our most stringent phylogeny-aware approach. We have now removed this discussion phrase from our revised manuscript; instead, the exact manner how the average proportion is calculated can be found with the appropriate context in the Results section.:

“We also applied the same reasoning to estimate how much divergence beyond recognition contributes to TRGs. To this aim we calculated the fraction of focal genes lacking detectable homologues in a phylogeny-based manner, in the target species and in all species more distantly related to the focal species than the target species (see Materials and methods and Figure 6—figure supplement 2A for a schematic explanation). Again, the observed proportion of TRGs far exceeded that estimated to have originated by divergence (the contribution of divergence ranging from 0% to 52% corresponding to the first and before-last “phylostratum” of the fly dataset tree respectively, with an overall average of 30%; Figure 6—figure supplement 2B and C).

Reviewer #1:

*This is important work that tries to estimate relative contribution of sequence divergence in the origin of novel genes (also known as orphan genes or taxonomically restricted genes). To my knowledge this is the first study entirely focused on this question which is essential for understanding which mechanism (divergence or* de novo *route) prevails in creating evolutionary innovations at the genome/proteome level. The authors devised a very neat protocol that detects conserved synteny regions between species and then looks for proteins that lack sequence similarity in the target genome but are otherwise embedded within these conserved synteny regions. By extrapolating the findings from the conserved synteny regions to the whole genome the authors conclude that divergence mechanism accounts for at most a third of eukaryotic novel genes.*

1) Subsection “Selecting optimal BLAST E-value cut-offs”, Figure 3A – It is really elegant how the authors estimated the proportion of false positive and false negative homologues. From their figures it is clear that the proportion of false positives changes quite differently from the proportion of false negatives depending on the e-value. The authors mention this in the text, but they don't discuss that controlling of false positives is more critical when deciding on the appropriate e-value cutoffs. This is evident from the shape of the Figure 3A curves, where false negatives show linear-like dependence and false positives exponential-like dependence, plus the false positive curves are not dependent on the evolutionary distance. In practice, this means that failing to properly control for false positives would generate spurious sequence similarity hits all over the place, whereas not perfectly adjusted e-value would not have a such profound effect on false negatives, especially in the phylogenetic context.

We thank the reviewer for this compliment and these suggestions. We have accordingly added the following descriptions and interpretation to our revised manuscript (section “Selecting optimal BLAST E-value cut-offs”):

“In Figure 3A, we can see how the ratios of undetectable and false homologies vary as a function of the BLAST E-value threshold used. […]This means that, when comparing relatively closely related species, failing to appropriately control for false homologies would have an overall more severe effect on homology detection than failing to account for false negatives.”

2) Linked to the previous comment: The authors discuss the BLAST "false negatives" debate but miss the balance by omitting to cite Domazet-Loso et al., 2017 paper which stresses the importance of evaluating BLAST "false positives". Given that the manuscript specifically deals with the false positives this paper should be cited.

We thank the reviewer for pointing out this omission. We have now mentioned the issue of false positives and cited this work in the relevant part of the section in p15 of our revised manuscript. The following phrase has been added:

“although the importance of spurious BLAST hits generating false positives has also been stressed (Domazet-Loso et al., 2017)”

3) Subsection “Calculation of undetectable and false homologies and definition of optimal E-values” – I had problems to understand how Mathews Correlation Coefficient was used to decide on the E-value cut-off. Could you please describe the protocol in more details here?

We apologize for the lack of clarity in this part of our submitted manuscript. The Mathews Correlation Coefficient was only used to calculate the general use E-value cut-offs found in Figure 3—figure supplement 1. The cut-offs used in the rest of the analyses in the article (0.01, 0.01, 0.001) were calculated by taking the highest E-value cut-off that keeps false positives under 5%. We nevertheless provide more details about the protocol relevant to this question in the Materials and methods section of our revised manuscript “Calculation of undetectable and false homologies and definition of optimal E-values”:

“We also calculated the Mathews Correlation Coefficient (MCC) measure of binary classification accuracy for every E-value cut-off. This is a balanced measure that takes into account true and false positives and negatives which can be used even in cases of extensive class imbalance. At every E-value cut-off, we treated undetectable homologies as False Negatives, and false homologies (matches to the reversed proteome) as False Positives. Similarly, sequence-detected homologies (defined based on micro-synteny) were treated as True Positives and genes for which the reversed-search gave no significant hit were treated as True Negatives. The MCC measure was then calculated at each E-value cut-off based on these four values using the *mcc* function of R package mltools. When multiple E-value cut-offs had the same MCC (rounded at the 3rd decimal), the highest (less stringent) E-value was retained. The results for each focal-target genome pair are shown in Figure 3—figure supplement 1 (“general E-value” column).”

4) Subsection “The rate of “divergence beyond recognition” and its contribution to thetotal pool of genes without similarity” – Presentation of the results in Figure 4B and 4C (Figure 6—figure supplement 2) is quite confusing.

We thank the reviewer for this comment. We have now clarified these figures by removing the B panels and making the axis labels more straightforward.

5) Subsection “Calculation of proportion of orphan genes due to processes other than sequence divergence” – "This is done by taking…" This part were phylogeny-based proportions are obtained was not comprehensible to me. Could you please make some schematic representation of this part of the protocol?

We apologize for the lack of clarity. A schematic representation has now been added to Figure 6—figure supplement 2 in our revised manuscript.

Reviewer #2:

*The authors have made a comprehensive attempt to quantify the contribution of sequence divergence as a source of orphan genes. They have ingeniously used micro-synteny to identify the orphan genes created by divergence. They estimate that at most one-third of the genes within the micro-syntenic regions result from sequence divergence. Subsequently, they extrapolate this result to the entire genome and conclude that the majority of orphan genes are not formed through sequence divergence. Although several studies have investigated orphan genes in various organisms, the relative contribution of processes such as* de novo *gene creation and sequence divergence in the formation of orphan genes remains largely unexplored. Hence, I consider that this work can significantly contribute to the progress of the field.*

Major concern:The authors assume that the similar distribution of evolutionary rates within and outside the syntenic blocks indicates that the proportion of orphan genes created through sequence divergence within the syntenic block can be extrapolated to the whole genome (Subsection “The rate of “divergence beyond recognition” and its contribution to the total pool of genes without similarity” paragraph 2). Given that the evolutionary rates cannot be estimated for the orphan genes lacking detectable homologs, the assumption made by the authors about the comparability of evolutionary rates across the genome is most likely based on a dataset that excludes such genes. Thus, their argument, although correct for the overall distribution of evolutionary rates, should not be used to extrapolate the proportion of orphan genes originated by divergence within the syntenic blocks to the whole genome.

We thank the reviewer for this insightful remark. Indeed, in our comparison of evolutionary rates inside and outside of syntenic blocks, orphan genes are not included, as it is impossible to do so. We wholeheartedly agree that is important to make this fact clear to the reader and explain its consequences for the conclusions of the article. Yet, while this is undoubtedly a caveat of our approach, it is crucial to understand that it is only a minor one.

We estimated the extent of the issue and found that the TRGs at the relevant phylogenetic levels only account for 0.0013, 0.008 and 0.029 of all genes in fruit fly, human and yeast respectively. This means that there is only a tiny fraction of the total information of genome-wide evolutionary rates not incorporated in the comparisons. Any effect of such a small percentage of genes can only be very limited. Yes, we cannot categorically reject the notion that, for this small fraction of genes, the same mutational processes that results in loss of synteny may also somehow result in more TRGs (although we show that they do not result in higher evolutionary rates in other genes). We therefore acknowledge this as a minor caveat of our methodology.

We now properly introduce this caveat when presenting the comparison of the evolutionary rates:

“It is impossible to directly compare the evolutionary rates of genes lacking homologues inside and outside conserved micro-synteny. However, such genes only account for a miniscule percentage of all genes in the genome: 0.0013, 0.008 and 0.029 in fly, human and yeast respectively. Despite these minimal caveats, evolutionary rates are globally very similar inside and outside regions of conserved micro-synteny, allowing to extrapolate with confidence”

We have also specified in all appropriate places in the revised manuscript that the extrapolation is approximative:

• In the schematic representation of the pipeline in Figure 2: “approximately extrapolated”.

• In the subsection “A synteny-based approach to establish homology beyond sequence similarity”: “This estimate can then be extrapolated genome-wide to approximate the extent of origin by complete divergence for orphan genes and TRGs outside of syntenic regions”.

• In the subsection “The rate of divergence beyond recognition and its contribution to the total pool of genes without similarity”: “can be used as a proxy for the genome-wide rate of origin-by-divergence for genes without detectable similarity”

• In the legend of Figure 6—figure supplement 2: “can be approximately extrapolated genome-wide”

In summary, thanks to the reviewer’s remark, we have taken great care to adjust the language around the extrapolation, to make evident in our revised manuscript that it is an approximation. We would like to stress that, since this is a novel solution to a hard, hitherto unsolved problem, this approximation still has much value despite certain caveats. Hence, we believe that, framed appropriately as in the revised manuscript, the extrapolation can and should be used.

Reviewer #3:[…] Providing positive evidence for the set of genes that have rapidly diverged is particularly important because it opens the path for researchers to explore the mechanisms whereby particular genes can evolve so much more quickly than the typical gene.The table with the abbreviations (e.g., ggor) should be moved from supplementary to the main text. That way the reader doesn't need to access supplementary to understand Figure 6.

We thank the reviewer for this remark. We have added this table to the main text in our resubmitted manuscript (Table 1) and find it much improves the interpretability if our figures.

In both yeast and Arabidopsis, ~50% of orphan genes are NOT located in (micro) syntenic regions of near relatives (Arendsee et al., 2019). These genes (and how they arose) are really interesting as well. The authors might want to mentioned this in the Discussion.

We thank the reviewer for this suggestion. We now mention this in the Discussion and cite the relevant article:

“We are able to isolate and examine these genes when they are found in conserved micro-synteny regions, but at this point we have only a statistical global view of the process of divergence outside of these regions. Since, for example, in yeast and in Arabidopsis, ~50% of orphan genes are located outside of syntenic regions of near relatives27, the study of their evolutionary origins represents exciting challenges for future work.

The manuscript mentions genes with "retention of structural similarity " that "suggest the possibility of conservation of ancestral signals in the absence of sequence similarity." Keeping at least some of a structure but losing homology!! Cool. How might this fit into evolution? or is it just a quirk?

We thank the reviewer for this interesting question. We do believe that this is a very intriguing question and that these cases where common Pfam matches are found but no pairwise similarity exists should be examined more closely in the future. However, another reviewer has pointed out that we have been a bit over-enthusiastic in our interpretation of this match and we should require further evidence of structural similarity; we have therefore edited this section to remove reference to the structural similarity idea. We hope that providing this evidence will be the focus of future investigations in the field.

The vocabulary the authors use (e.g., "twilight zone", "freefall") adds to the paper.

We appreciate the reviewer’s positive view of these terms.

[Editors’ note: what follows is the authors’ response to the second round of review.]

Reviewer #2:[…] The authors accept that the proportion of orphan genes within the syntenic blocks decreases as the criterion is made more stringent. They suggest this is because a lesser number of genes were found within the syntenic regions (Rebuttal: Reviewer 2, Minor comment 3), but the indicated change in the proportion of orphan genes suggests that as more genes are included within syntenic blocks, the proportion of orphan genes within the blocks rises. This is quite intuitive because in many pairwise comparisons (19/48) the number of "match not found" within syntenic blocks is in single-digit and can potentially increase many folds by the mere addition of few more genes (Supplementary Table: Figure 3—figure supplement 1).Thus, with more relaxed criteria, the calculated divergence contribution will increase as higher fractions of orphan genes are included within syntenic blocks, but the total fraction of orphan genes does not change. Here, the compatibility of the evolutionary rates within and outside the blocks can still be maintained, while the calculated contribution made by divergence will vary.In the rebuttal, the authors write:"Note that, although, as expected, stricter synteny criteria led to fewer genes being found in conserved micro-syntenic blocks, overall results changed minimally between the two versions and hence can be considered robust."It has to be clearly shown, that increasing the stringency of the synteny criterion does not specifically exclude "match not found" genes and their proportion within the syntenic blocks is not a function of this criterion. The authors are advised to make this clarification within the manuscript before making genome level extrapolation. The method established by the authors is robust, and certainly, it will be extensively used in the future. However, the formula used for the genome-level extrapolation appears extremely sensitive to the synteny criterion, unless shown otherwise. Given that the authors have data supporting that synteny criterion does not affect the overall result, they should show it.

We thank the reviewer for their in-depth analysis of this aspect of our work. Clearly, the reviewer has perfectly understood the methodological details and has laid out a potential problem that could affect the main finding of our manuscript. We wholeheartedly agree that it is important that the reader and any future adopters of this method understand the relationship of the synteny criterion to the proportion explained by divergence. In order to be as exhaustive as possible, we performed additional analyses using one more relaxed and one more stringent conserved synteny definition. Our results clearly show that relaxing the synteny definition compared to the one we used in the study has only minimal impact on the proportion explained by divergence, despite allowing some known false positives and raising the number of conserved synteny regions almost by a factor of two. Using the even more stringent definition was problematic because in many cases the number of identified regions was too low to allow further analyses. Hence such a definition would be impractical for our study. Nevertheless, analysis of a subset of cases where the number of identified regions was sufficient, showed a larger impact than in the previous comparison (relaxed vs. the one used in the study) which was nonetheless still not dramatic.

We now mention these results in our Results section and include details in the legend of a new supplementary figure:

**“**We extrapolated the proportion of genes without detectable similarity that have originated by complete divergence, as calculated from conserved micro-synteny blocks (Figure 4), to all genes without similarity in the genome (Figure 6, see Materials and methods and Figure 6—figure supplement 1 for detailed description). […] This implies a substantial role for other evolutionary mechanisms such as de novoemergence and horizontal gene transfer, although horizontal gene transfer is not known to be frequent in metazoa.”

We also provide methodological details in Materials and methods:

The specific choice of synteny criterion was made after we conducted an initial trial with a minimum of one homologue on either side, which showed limited false positives, revealed by visual inspection (obvious cases of non-homologous genes which, due to rearrangements such as micro-inversions were placed “opposite” each other, even though they did not originate by divergence). […] The stringent definition added an additional syntenic homologue to either side relative to the current one: three genes on either side need to have syntenic homologues that are separated by one gene at most and, again, we retain cases where the outer neighbors (-3,+3) have no homologues at all to the target genome.”